# Reactivation of RNA metabolism underlies somatic restoration after adult reproductive diapause in *C. elegans*

Nikolay Burnaevskiy[1], Shengying Chen[1], Miguel Mailig[1], Anthony Reynolds[1], Shruti Karanth[1], Alexander Mendenhall[1], Marc Van Gilst[2], Matt Kaeberlein[1]*

[1]Department of Pathology, University of Washington, Seattle, United States;
[2]Department of Anesthesiology and Pain Medicine, University of Washington, Seattle, United States

**Abstract** The mechanisms underlying biological aging are becoming recognized as therapeutic targets to delay the onset of multiple age-related morbidities. Even greater health benefits can potentially be achieved by halting or reversing age-associated changes. *C. elegans* restore their tissues and normal longevity upon exit from prolonged adult reproductive diapause, but the mechanisms underlying this phenomenon remain unknown. Here, we focused on the mechanisms controlling recovery from adult diapause. Here, we show that functional improvement of post-mitotic somatic tissues does not require germline signaling, germline stem cells, or replication of nuclear or mitochondrial DNA. Instead a large expansion of the somatic RNA pool is necessary for restoration of youthful function and longevity. Treating animals with the drug 5-fluoro-2′-deoxyuridine prevents this restoration by blocking reactivation of RNA metabolism. These observations define a critical early step during exit from adult reproductive diapause that is required for somatic rejuvenation of an adult metazoan animal.
DOI: https://doi.org/10.7554/eLife.36194.001

*For correspondence:
kaeber@uw.edu

## Introduction

While many interventions that delay age-associated declines have been described in laboratory studies, instances of pausing or even reversing aging are relatively rare and generally do not preserve or restore youthful function in all tissues of an animal (*Kaeberlein et al., 2015*). Diapause states have been discovered in *C. elegans* that apparently preserve or restore functionality of chronologically old tissues after diapause exit. One example of such a state is the adult reproductive diapause (ARD), which is induced in developmentally mature animals by starvation. The animals show signs of tissue and cellular aging over a period of several weeks, but become apparently fully rejuvenated and proceed to have a normal adult lifespan after exit from ARD (*Angelo and Van Gilst, 2009*).

Depending on the developmental point at which *C. elegans* face starvation, animals can enter diapause at either the first larval stage L1 (L1 arrest), the second larval stage L2 (dauer), or, in the case of ARD, just after the transition from the fourth larval stage (L4) to young adulthood (*Angelo and Van Gilst, 2009*; *Johnson et al., 1984*; *Klass and Hirsh, 1976*). Developmental arrests at the third and fourth larval stages (L3 and L4) have also been recently described, but are less studied (*Schindler et al., 2014*). Dauer is the most studied diapause in worms and represents a clearly distinct, alternative third larval stage that is highly stress resistant and long-lived (*Klass and Hirsh, 1976*; *Zhou et al., 2011*; *Fielenbach and Antebi, 2008*). Dauer worms undergo significant anatomical transformations, such as synthesis of the specialized cuticle and mouth plug (*Frézal and Félix, 2015*; *Hu, 2007*). These morphological changes make it difficult to investigate aging at the cellular

and molecular level in dauer animals, although animals that have exited dauer after a prolonged period of time have a normal adult lifespan (*Fielenbach and Antebi, 2008*).

Unlike dauer, ARD does not require major anatomical transformations and therefore may provide direct insight into the mechanisms that preserve functionality of cells and tissues during prolonged diapause and restore them upon the diapause exit. Entry into ARD occurs when animals face starvation at the transition from the L4 stage of development to adulthood (*Angelo and Van Gilst, 2009*). During ARD, developmentally mature animals show signs of tissue and cellular aging along with reduction of the germline, atrophy of intestine and somatic gonad, and appearance of dead embryos in the uterus (*Angelo and Van Gilst, 2009*). Careful analysis by the Kimble group suggested that the germline is sacrificed for production of a small number of progeny during ARD (*Seidel and Kimble, 2011*). Exit from ARD is marked by drastic morphological improvements: animals resume growth, germline stem cells (GSCs) repopulate the germline, and the atrophied intestine and gonad regain their youthful appearance (*Angelo and Van Gilst, 2009*). Post-ARD animals appear to be fully rejuvenated. In addition, animals exiting from ARD have a normal adult lifespan and are capable of progeny production despite their time as diapaused adults. Intriguingly, the observed morphological restoration takes place in fully developed adult animals in which all somatic cells are post-mitotic. Although ARD was discovered nearly a decade ago, the signals and molecular mediators of ARD maintenance and exit remain unknown.

In this study, we examined molecular and physiological mechanisms for post-ARD tissue restoration. We found that the DNA synthesis inhibitor 5-fluoro-2'-deoxyuridine (FUDR) prevents normal recovery from ARD. Surprisingly, neither germline nor somatic DNA synthesis were required for post-ARD tissue recovery. Instead, recovery of somatic tissues involves nucleolar reactivation and a large expansion of the RNA pool, both of which are sensitive to FUDR. These findings provide insight into a key early steps in post-ARD somatic restoration, and allow better understanding of the pleiotropic effects of FUDR, including effects on longevity, that have been observed by many laboratories in recent years (*Van Raamsdonk and Hekimi, 2011*; *Aitlhadj and Stürzenbaum, 2010*; *García-González et al., 2017*; *Kato et al., 2017*; *Anderson et al., 2016*; *Angeli et al., 2013*).

## Results

### FUDR prevents tissue restoration upon exit from ARD

Transition into and exit from the ARD induce dramatic morphological changes. Specifically, entry into the diapause state is marked by cessation of growth, shrinkage of the intestine and somatic gonad, and reduction of the germline (*Figure 1A,B,D*). Upon exit from ARD growth resumes, the germline repopulates, and the intestine and gonad regain normal appearance (*Figure 1A,B,D*). Importantly, in addition to visual improvements, animals regain youthful levels of functional health measures, such as motility and pharynx pumping (food grinding) (*Figure 1E,F*). Therefore, exit from ARD induces functional and morphological restoration to a youthful state. We will use the term 'post-ARD' hereafter to refer to the animals exiting from the diapause. In contrast, genetically matched animals that were never starved are referred to as 'control'.

To dissect the mechanisms of post-ARD recovery, we first tested the model that signals from the germline to promote restoration of other tissues. The germline is a potent regulator of *C. elegans* longevity (*Arantes-Oliveira et al., 2002*). Regrowth of this tissue from a small pool of preserved stem cells is the most drastic morphological improvement associated with the diapause exit. We therefore hypothesized that the signals arising from the growing germline stimulate morphological restoration of somatic tissues as well. To test this model, we blocked germline re-growth with FUDR, a DNA synthesis inhibitor that is commonly used in *C. elegans* to chemically sterilize adult animals (*Gandhi et al., 1980*).

We transferred either control L4 larvae or animals that had been maintained in ARD for 3 weeks onto fresh plates containing abundant *E. coli* OP50 food with or without 50 μM FUDR. While ARD animals recovered well in the absence of the drug, FUDR completely prevented germline regrowth or restoration of youthful morphology and functionality (*Figure 1A,B,D,E,F*). In addition, dead embryos were still present in the uterus of post-ARD animals on FUDR (*Figure 1A*, arrows). In contrast, FUDR did not have a strong effect on control L4 larvae, which progressed into adulthood

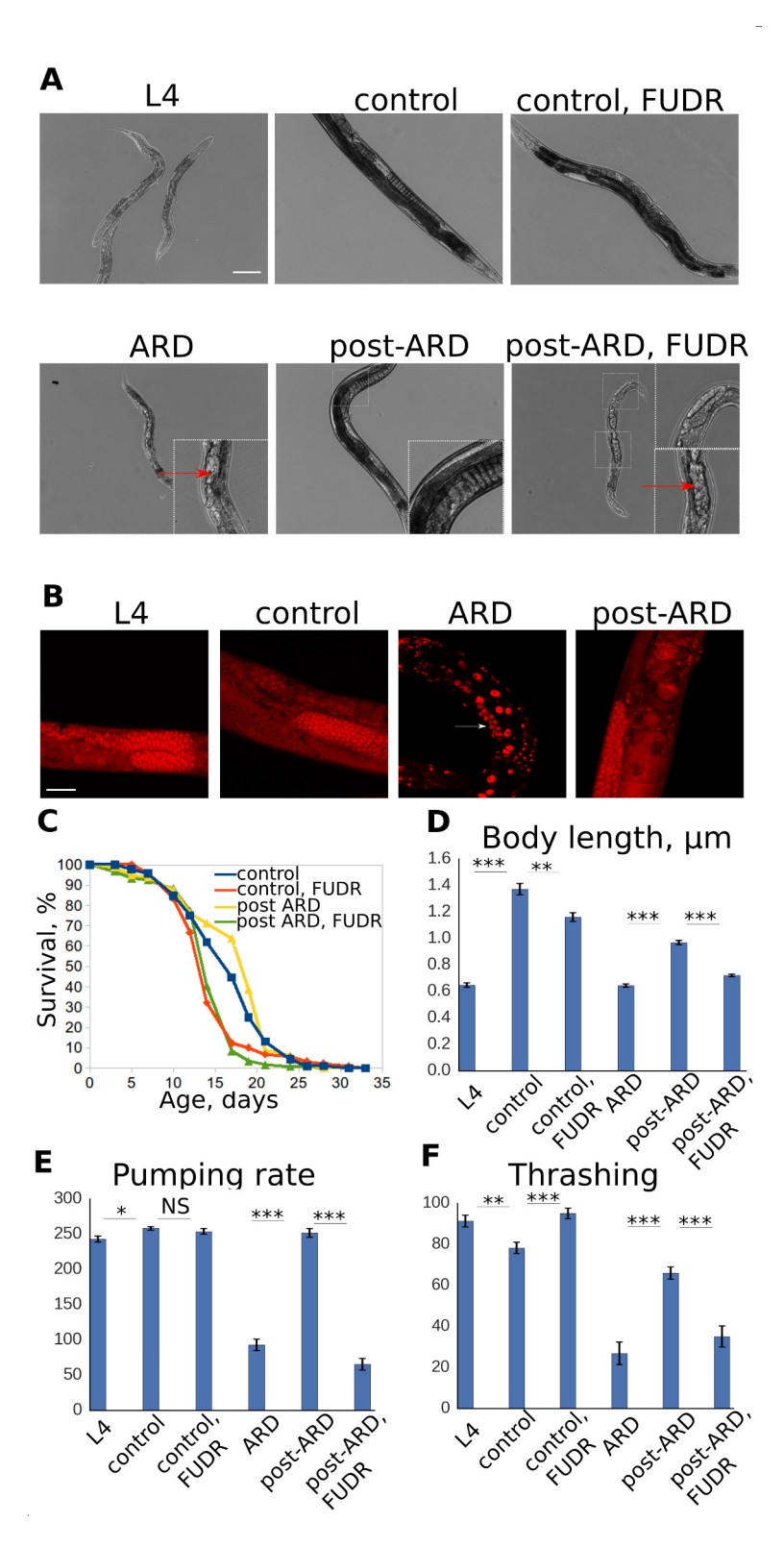

**Figure 1.** FUDR prevents post-ARD recovery. (**A**) Representative DIC images of control, ARD and post-ARD animals. Upper panel: control L4 larva, 4 day old control adult, and 4 day old adult treated with FUDR from L4; lower panel: animal after 3 weeks in ARD (ARD), and animals which were allowed to recover from ARD for 4 days with and without FUDR (post-ARD). Insets show dead embryos and non-recovered germline in post-ARD FUDR condition. Scale bar is 100 μm. (**B**) DAPI staining of germline in L4 larva, 4 day old adult (control), animal that spent 3 weeks in diapause (ARD), and

*Figure 1 continued on next page*

*Figure 1 continued*

animal after 4 days of recovery (post-ARD). Reduced pool of GSCs in ARD animal is indicated by an arrow. Scale bar is 25 µm. (**C**) Survival curves of the control and post-ARD animals with or without FUDR. On day 0 of the experiment, ARD animals after 3 weeks in diapause and control L4 were transferred onto fresh OP50-seeded plates. The next day was considered day1 of adulthood for both conditions. See *Figure 1—source data 1*. (**D**) Body length (in micrometers) of L4 larvae, ARD, 4 day old control and post-ARD animals after 4 days of recovery. Shown are mean values ± SEM. See *Figure 1—source data 2*. (**E**) Pumping rates of control and post-ARD animals. L4 and 3 weeks old ARD animals were scored about 1 hr after transfer to fresh OP50 plates. The same cohorts (control and post-ARD) were examined again 4 days later. Indicated are rates per minute. Shown are mean values ± SEM. See *Figure 1—source data 3*. (**F**) Thrashing rates (per minute) of control and post-ARD animals. Shown are mean values ± SEM. See *Figure 1—source data 4*. *p<0.05; **p<0.005; ***p<0.0005.

DOI: https://doi.org/10.7554/eLife.36194.002

The following source data and figure supplement are available for figure 1:

**Source data 1.** N2 lifespans.
DOI: https://doi.org/10.7554/eLife.36194.004
**Source data 2.** Body length.
DOI: https://doi.org/10.7554/eLife.36194.005
**Source data 3.** N2 pumping.
DOI: https://doi.org/10.7554/eLife.36194.006
**Source data 4.** N2 thrashing.
DOI: https://doi.org/10.7554/eLife.36194.007
**Figure supplement 1.** Representative DIC images of control and post-ARD animals treated with hydroxyurea.
DOI: https://doi.org/10.7554/eLife.36194.003

(*Figure 1A*). We used body size increase, disappearance of dead embryos from the uterus, and overall visual improvements to score post-ARD recovery rates in subsequent experiments.

To test if post-ARD animals are more sensitive to chemical stresses in general, we analyzed recovery of post-ARD animals in the presence of hydroxyurea (HU). HU has a broad cytotoxic effect by acting on multiple targets, including reducing the availability of deoxynucleotides through inhibition of ribonucleotide reductase (*Singh and Xu, 2016*; *Singh et al., 2017*). While post-ARD animals showed only mild changes during morphological recovery in the presence of HU, control animals experienced rapid demise and collapse of body morphology within 4 days (*Figure 1—figure supplement 1*). Therefore, the inhibitory effect of FUDR on post-ARD animals is not due to a general increase in sensitivity to chemicals.

We then tested how FUDR affects longevity of animals following exit from ARD. The drug is routinely used in lifespan experiments to prevent hatching of progeny and is typically applied at late L4 stage (*Gandhi et al., 1980*; *Sutphin and Kaeberlein, 2009*). To compare the effect of FUDR on young adult and ARD animals, we transferred animals to OP50 feeding plates prepared either with or without FUDR at either mid-L4 (when we would initiate ARD) or 3 weeks after entry into ARD. Despite the pronounced phenotypic difference produced by the drug following exit from ARD, the lifespans of both control and post-ARD animals on FUDR were not significantly different (*Figure 1C*). Therefore, FUDR prevents morphological restoration following exit from ARD without significantly impacting restoration of normal adult longevity, potentially uncoupling healthspan from lifespan.

## The germline is dispensable for ARD entry and recovery

To further test whether the effects of FUDR on recovery from ARD are due to inhibition of germline DNA synthesis, we used a worm strain carrying temperature-sensitive mutation in *glp-1*, which encodes a Notch receptor homolog essential for maintenance of the undifferentiated stem cells in the germline (*Austin and Kimble, 1987*). At the restrictive temperature, *glp-1(e2141)* animals develop into sterile adults that are virtually devoid of GSCs. If GSCs play an essential role in ARD, then *glp-1* mutants grown at the restrictive temperature should be unable to maintain or exit from the diapause state, and FUDR should not further impact *glp-1(e2141)* animals. Wild type N2 and *glp-1(e2141)* animals were raised at the restrictive temperature (25°C) and subjected to starvation at the L4 stage to induce ARD, which occurred normally in both genetic backgrounds. When animals were returned to fresh feeding plates to induce exit from ARD, both N2 and *glp-1(e2141)* animals resumed their growth, regained a youthful appearance, and restored pumping rate comparable to controls,

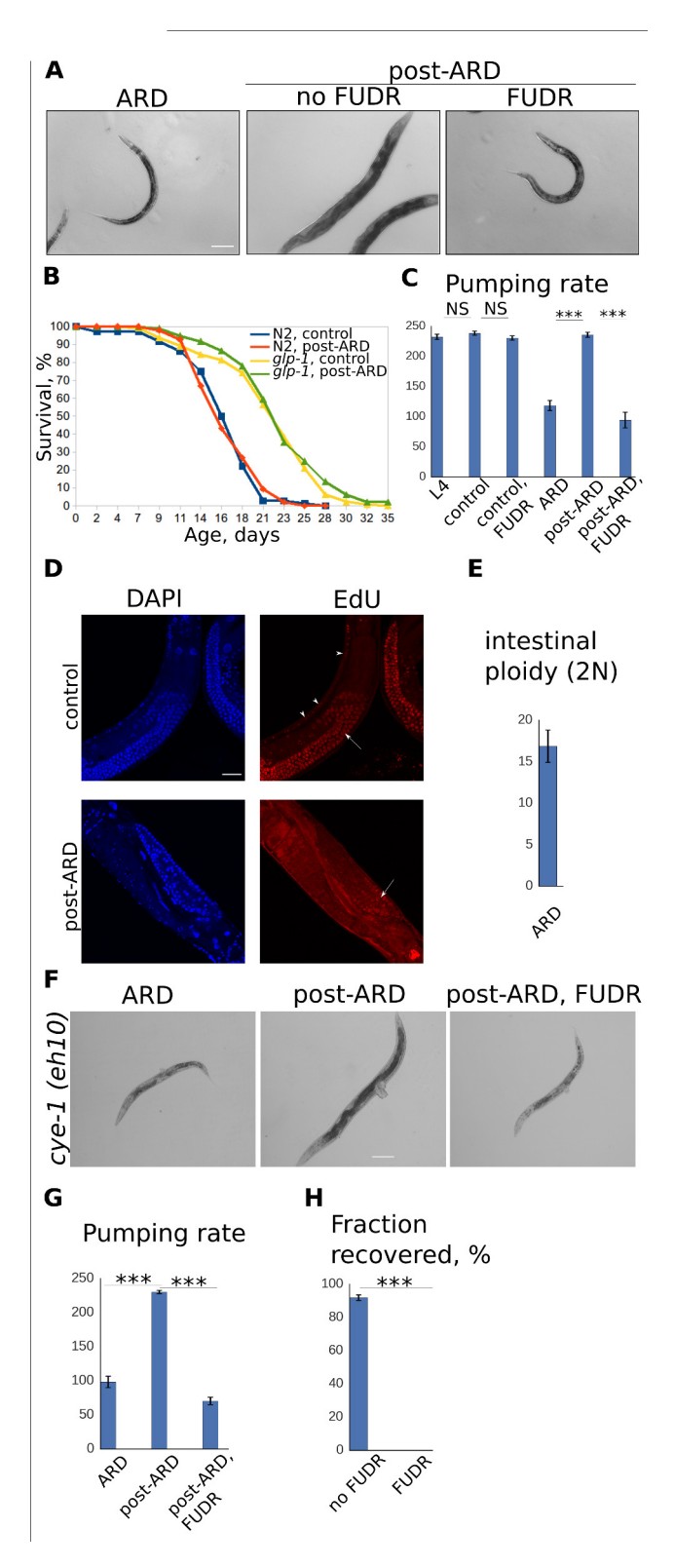

**Figure 2.** Adult reproductive diapause does not require germline or cell cycle activation. (**A**) DIC images of ARD and post-ARD *glp-1 (e2141)* animals. FUDR prevented morphological improvements of the sterile *glp-1* mutants. Scale bar is 100 μm. (**B**) Survival curves of the wild type (WT) and *glp-1* mutants, control and post-ARD. On day 0, two weeks old ARD animals and control L4 were transferred onto fresh OP50. The next day was considered day1 of adulthood for both conditions. Shown are survival plots of recovered animals only. See *Figure 2—source data 1*. Also see *Figure 2—figure*

*Figure 2 continued on next page*

*Figure 2 continued*

*supplement 1*. (C) Pumping rates (per minute) for control and post-ARD *glp-1 (e2141)* animals. Shown are mean values ± SEM. See *Figure 2—source data 2*. (D) EdU labeling of control and post-ARD animals. Control animals were grown on EdU-labeled bacterial cells from egg till day 2 of adulthood. ARD animals were transferred on EdU-labeled cells for recovery and examined 5 days later. Arrows point to EdU-labeled germline cells nuclei, arrowheads point to EdU-labeled somatic nuclei. No somatic cells labeling was observed in post-ARD animals. Scale bar is 25 μm. (E) Ploidy (in diploid equivalents) of intestinal cells in ARD animals. Shown is the mean value ± SEM. See *Figure 2—source data 3*. (F) DIC images of ARD and post-ARD *cye-1 (eh10)* animals. FUDR prevented morphological improvements of *cye-1* mutants. Scale bar is 100 μm. (G) Pumping rates (per minute) of control and post-ARD *cye-1 (eh10)* animals. Shown are mean values ± SEM. See *Figure 2—source data 4*. (H) Fraction of *cye-1 (eh10)* animals that morphologically recovered after diapause with and without FUDR. Shown are mean values ± SEM. See *Figure 2—source data 5*. Also see *Figure 2—source data 6*. *p<0.05; **p<0.005; ***p<0.0005.
DOI: https://doi.org/10.7554/eLife.36194.008

The following source data and figure supplement are available for figure 2:

**Source data 1.** N2 and glp-1 lifespans.
DOI: https://doi.org/10.7554/eLife.36194.010
**Source data 2.** glp-1 pumping rate.
DOI: https://doi.org/10.7554/eLife.36194.011
**Source data 3.** Intestinal ploidy.
DOI: https://doi.org/10.7554/eLife.36194.012
**Source data 4.** cye-1 pumping.
DOI: https://doi.org/10.7554/eLife.36194.013
**Source data 5.** cye-1 recovery rate.
DOI: https://doi.org/10.7554/eLife.36194.014
**Source data 6.** N2 and glp-1 lifespans.
DOI: https://doi.org/10.7554/eLife.36194.015
**Figure supplement 1.** Survival curves of the wild type (WT) and *glp-1* mutants, control and post-ARD.
DOI: https://doi.org/10.7554/eLife.36194.009

indicating normal recovery (*Figure 2A,C*). Furthermore, post-ARD lifespans of the reproductively active N2 and sterile *glp-1(e2141)* animals after two weeks in ARD did not differ significantly from adult lifespans of the genetically matched controls that never experienced ARD (*Figure 2B*, *Figure 2—figure supplement 1*). While we cannot rule out the possibility that *glp-1(e2141)* mutants may still have a small number of GSCs at the restrictive temperature, any remaining GSCs clearly fail to repopulate the germline. Therefore, we conclude that restoration of the germline is not required for the proper maintenance of, or recovery from ARD, and we find no evidence to support our prior model that signals from GSCs preserved during ARD mediate morphological and functional restoration and maintenance of full lifespan potential.

Although the germline is not necessary for ARD, it is still possible that FUDR blocks post-ARD recovery via its effect on the germline, for example by inducing a cell non-autonomous genotoxic stress signal in the germline that prevents morphological and functional restoration of somatic tissues. We therefore compared recovery of the sterile *glp-1(e2141)* animals with and without FUDR. As in wild type N2 animals, FUDR blocked the morphological and functional improvements upon exit from ARD in *glp-1(e2141)* animals (*Figure 2A,C*). Thus, we conclude that the effects of FUDR on ARD are unlikely to involve the germline or GSCs.

## ARD recovery does not require activation of the cell cycle in somatic tissues

Since the inhibitory effect of FUDR was not due to its effect on the germline, we reasoned that post-ARD recovery may require activation of the cell cycle in somatic cells. For instance, it is possible that during progression from L4 to adulthood in the absence of nutrients, some somatic cell divisions or endoreduplications are postponed in ARD animals. In this scenario, completion of the skipped DNA replication cycles may be required for normal post-ARD recovery. To test this hypothesis, we examined synthesis of new DNA through metabolic labeling with 5-ethynyl-2'-deoxyuridine (EdU). When control animals are grown from egg to adulthood on EdU-labeled bacteria, birth of new DNA is observed both in germline and in somatic cells (*Figure 2D*). In contrast, EdU labeled only the

germline in post-ARD animals (*Figure 2D*), suggesting that little or no additional DNA replication occurs in somatic tissues of post-ARD animals.

*C. elegans* are known to have polyploid tissues. For instance, intestinal cells normally undergo four rounds of endoreduplications with a final round of DNA synthesis during the molt from L4 to adulthood (*Hedgecock and White, 1985*). We were interested to examine if final intestinal endoreduplication is postponed during the diapause and only resumes during post-ARD recovery. However, we were unable to detect endoreduplication with EdU even in control animals grown on EdU-labeled bacteria from egg (*Figure 2D*), indicating that this method is not suitable for analysis of ploidy changes. Instead, to determine if intestinal cells need to undergo a round of endoreduplication upon post-ARD recovery, we estimated ploidy of intestinal cells in ARD animals using DAPI staining. We found that intestinal nuclei in ARD animals contain approximately 16 equivalents of diploid genomes, indicating that these cells have already completed endoreduplications prior to exit from ARD (*Figure 2E*).

To directly test whether cell cycle activation in somatic tissues is required during exit from ARD, we analyzed post-ARD recovery of cyclin E-deficient animals and their sensitivity to FUDR. *cye-1 (eh10)* animals are maintained as a heterozygous strain with a balancer chromosome. Homozygous *cye-1(eh10)* mutants are viable, but do not complete endoreduplications, have smaller body size and defects in the development of reproductive system (*Lozano et al., 2006*). We found that somatic tissues of homozygous *cye-1 (eh10)* animals recover from ARD normally and that this recovery is blocked by FUDR (*Figure 2F,G,H*). Therefore, reactivation of cell cycle (or endocycle) is not required for post-ARD recovery.

## Mitochondrial DNA replication is not required for recovery from ARD

Neither germline nor somatic nuclear DNA synthesis were required for post-ARD recovery, hence we speculated that inhibition of somatic mitochondrial DNA replication might underlie the inhibitory effects of FUDR in this context. To test this possibility, we utilized a strain homozygous for a loss of function allele of the mitochondrial DNA polymerase gene *polg-1*. Homozygous *polg-1(ok1548)* mutants are viable and develop into morphologically normal adults, but are sterile and only produce a few inviable progeny (*Bratic et al., 2009*). The strain is maintained with a balancer chromosome containing a wild type copy of *polg-1*. For these experiments, we used a strain in which the balancer chromosome carries the fluorescent marker *myo-2::GFP* expressed in the pharynx. We compared post-ARD recovery in *polg-1(ok1548/+)* heterozygotes and *polg-1(ok1548)* siblings that can be distinguished by the presence of the fluorescent marker (*Figure 3—figure supplement 1A*). Consistent with the requirement for mitochondrial DNA synthesis for germline formation, homozygous *polg-1* mutants did not re-grow their germline after recovery from ARD (*Figure 3A*, arrow). However, we did not observe any other morphological defects in post-ARD *polg-1* mutants when compared to heterozygous siblings (*Figure 3A*). Similar fractions of *polg-1(ok1548/+)* and *polg-1(ok1548)* animals recovered in all the experiments (*Figure 3D*). In addition, we examined mitochondrial DNA content in ARD and post-ARD animals using qPCR. We again utilized sterile *glp-1(e2141)* animals to only analyze somatic cells. As shown in *Figure 3—figure supplement 1B*, we did not detect any increase of mitochondrial DNA content during post-ARD recovery. Therefore, synthesis of new mitochondrial DNA is not required for somatic recovery after ARD.

## RNA, but not DNA, metabolism is required for recovery from ARD

Neither germline nor somatic DNA synthesis could explain the ability of FUDR to block post-ARD recovery. We therefore considered the alternative possibility that FUDR acts on post-ARD recovery by inhibiting RNA metabolism. While both FUDR and the related compound 5-fluorouracil (5-FU) were long thought to only inhibit DNA synthesis, recent evidence suggests an RNA-targeted mechanism of toxicity of 5-FU (*García-González et al., 2017*; *Burger et al., 2010*; *Hoskins and Butler, 2008*; *Hoskins and Scott Butler, 2007*). We therefore tested whether supplementation of extra DNA or RNA nucleotides could suppress FUDR toxicity in the context of ARD. Uracil, but not thymine, largely rescued post-ARD recovery in the presence of FUDR (*Figure 3B*), with a majority of FUDR-treated animals able to undergo recovery upon uracil supplementation (*Figure 3E*). Similarly, the inhibitory effect of FUDR was suppressed by uridine, but not thymidine (*Figure 3—figure supplement 1C,D*). Deoxyuridine also rescued post-ARD recovery (*Figure 3—figure supplement 1C,*

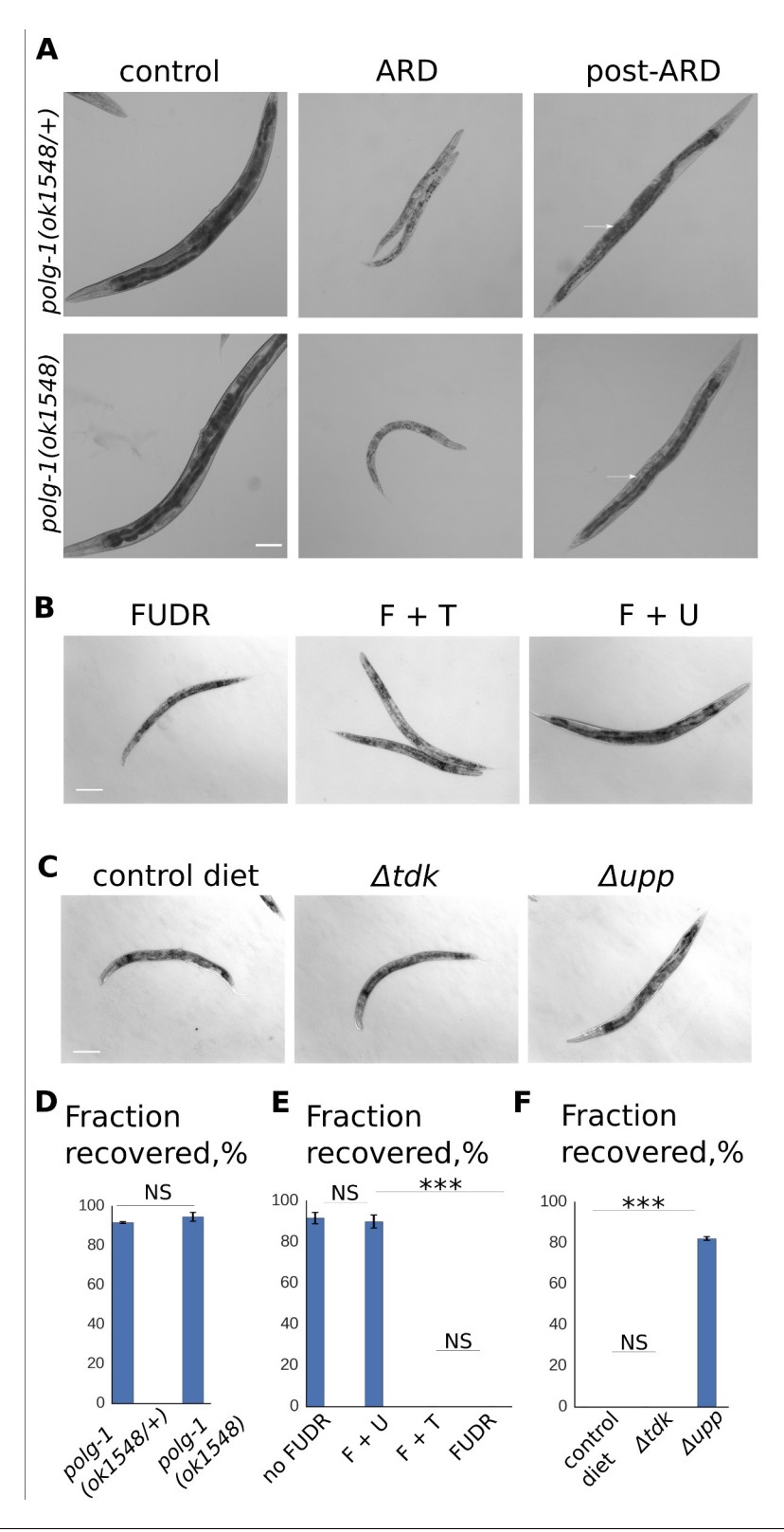

**Figure 3.** RNA, but not DNA, metabolism is required for post-ARD recovery. (**A**) DIC images of *polg-1 (ok1548/+)* heterozygotes and *polg-1 (ok1548)* mutants. Animals spent 3 weeks in the diapause and were then allowed to recover for 4 days on fresh OP50 plates. Arrows point to newly produced eggs in case of *polg-1 (ok1548/+)* heterozygotes and to non-recovered germline in case of *polg-1 (ok1548)* mutants. Scale bar is 100 µm. (**B**) DIC images of post-ARD animals recovering on 50 µM FUDR, 50 µM FUDR +1 mM uracil and 50 µM FUDR +1 mM thymine. Uracil, but not thymine rescued

*Figure 3 continued on next page*

*Figure 3 continued*

growth and recovery of post-ARD animals in presence of FUDR. Scale bar is 100 µm. (**C**) DIC images of post-ARD animals recovering on NGM plates containing 25 µM FUDR and seeded with different bacterial cells from Keio knockout collection. Control diet – BW25113, Δ*tdk* – JW1226-1, Δ*upp* – JW2483-1. Inhibitory effect of FUDR was reduced by JW2483-1 cells. Scale bar is 100 µm. (**D**) Fraction of *polg-1 (ok1548/+)* and *polg-1 (ok1548)* animals that morphologically recovered after ARD. Shown are mean values ± SEM. See *Figure 3—source data 1*. (**E**) Fractions of animals that morphologically recovered after ARD in various conditions (see *Figure 3B*). Shown are mean values ± SEM. See *Figure 3—source data 2*. (**G**) Fractions of animals that morphologically recovered after ARD with different bacterial diets in presence of FUDR (see *Figure 3C*). Shown are mean values ± SEM. See *Figure 3— source data 3*. *p<0.05; **p<0.005; ***p<0.0005.

DOI: https://doi.org/10.7554/eLife.36194.016

The following source data and figure supplements are available for figure 3:

**Source data 1.** polg-1 recovery rate.
DOI: https://doi.org/10.7554/eLife.36194.019
**Source data 2.** Recovery with uracil.
DOI: https://doi.org/10.7554/eLife.36194.020
**Source data 3.** Recovery with bacterial mutants.
DOI: https://doi.org/10.7554/eLife.36194.021
**Source data 4.** mtDNA content.
DOI: https://doi.org/10.7554/eLife.36194.022
**Source data 5.** Recovery with uridine.
DOI: https://doi.org/10.7554/eLife.36194.023
**Figure supplement 1.** DNA metabolism is dispensable for post-ARD recovery.
DOI: https://doi.org/10.7554/eLife.36194.017
**Figure supplement 2.** Post-ARD wild type animals and *tyms-1* mutants are shown.
DOI: https://doi.org/10.7554/eLife.36194.018

**D**); however, interpretation of this experiment is difficult due to the fact that deoxyuridine may compete with FUDR (5-fluoro-2'-deoxyuridine) for the same targets and interfere with FUDR metabolism in the animal.

Recent reports indicated that bacterial metabolism affects the response to FUDR in *C. elegans* (*García-González et al., 2017*; *Scott et al., 2017*). To further distinguish between requirements for RNA versus DNA metabolism in recovery of post-ARD animals, we utilized bacterial mutant strains deficient in nucleotide-processing enzymes. Namely, we used strains JW1226-1 (deficient in thymidine kinase, Δ*tdk*), JW2483-1 (deficient in uracil phosphoribosyltransferase, Δ*upp*) and BW25113 (parent strain, control diet) from the Keio collection. We expected FUDR toxicity to be reduced by JW2483-1 compared to parent strain BW25113, due to the inability of this strain to efficiently incorporate FUDR derivatives into RNA metabolism. In contrast, we expected that loss of *tdk* would not affect sensitivity of post-ARD animals to FUDR. As shown in *Figure 3C,F* the inhibitory effect of FUDR was indeed reduced by JW2483-1 (Δ*upp*), but not JW1226-1 (Δ*tdk*) compared to the parent strain BW25113.

In addition, we examined a worm strain carrying a temperature sensitive allele of thymidine synthase *tyms-1(hc65)*. TYMS-1 is one of the targets of FUDR that mediates its DNA-based toxicity (*Wyatt and Wilson, 2009*; *Santi et al., 1974*). Animals homozygous for the *tyms-1(hc65)* allele recovered normally from ARD at the restrictive temperature (*Figure 3—figure supplement 2*). Taken together, these observations support the model that FUDR prevents post-ARD recovery through inhibition of RNA metabolism.

## Nucleolar expansion is an early step in post-ARD recovery

5-FU exerts its anti-cancer activity at least in part through inhibition of ribosomal RNA (rRNA) production and induction of ribosomal stress (*Burger et al., 2010*; *Fang et al., 2004*; *Sloan et al., 2013*; *Sun et al., 2007*). Therefore, the strong inhibitory effect of FUDR on post-ARD animals may result at least partially from suppression of rRNA biogenesis. To examine the impact of ARD and FUDR on rRNA metabolism, we first examined nucleolar morphology (*Uno et al., 2013*). Synthesis and processing of rRNA occur in the nucleolus, a phase separated organelle inside the nucleus, which is organized around rDNA repeats encoding the rRNA. To track nucleolar status, we used a

transgenic strain producing GFP-tagged homologue of fibrillarin (FIB-1), a nucleolar methyltransferase involved in post-transcriptional rRNA processing.

Examination of ARD animals revealed that their nucleoli were dramatically smaller than in control L4 larvae or young adults (*Figure 4A,C*). When animals exited ARD, their nucleoli rapidly increased in size and reached a size similar to that of young, well-fed controls. FUDR did not prevent the nucleolar expansion upon exit from ARD (*Figure 4A,C*, *Figure 4—figure supplement 1*), suggesting that early steps of nucleolar reactivation may not be impaired by the drug. This interpretation is complicated somewhat by the unexpected finding that FUDR caused a pronounced expansion of nucleoli in control animals that had never experienced ARD (*Figure 4A,C*). These effects were independent of the germline (*Figure 4B,D*) and were reduced by addition of uracil to the growth medium (*Figure 4A,C*). Thus, we conclude that FUDR affects nucleolar structure in somatic post-mitotic tissues of wild type animals.

## FUDR impairs post-ARD expansion of the cellular RNA pool and rRNA processing

To better understand the changes in cellular RNA landscape associated with post-ARD recovery and FUDR treatment, we analyzed total RNA, as well as rRNA, in control and post-ARD animals. To focus only on somatic tissues, we used sterile *glp-1(e2141)* worms grown at the restrictive temperature. Total RNA was extracted from post-ARD animals and analyzed on an Agilent TapeStation system (*Figure 5A*). It was immediately apparent that FUDR impairs expansion of total RNA upon exit from ARD, as approximately 3.5-fold more FUDR-treated animals were needed to obtain comparable loading on the gel. Quantitatively, we found that animals not treated with FUDR expand their RNA content around 20-fold during exit from ARD, while FUDR almost completely prevented this RNA pool expansion (*Figure 5B*). This defect was mostly suppressed by uracil supplementation (*Figure 5B*). As shown in *Figure 5—figure supplement 1*, the overall RNA landscape looked similar in all conditions, pointing to strong inhibition of all RNA species by FUDR. Based on these observations, we propose that a strong inhibitory effect of the drug on RNA metabolism underlies its effect on post-ARD recovery.

We also specifically examined changes to rRNA in these conditions. FUDR caused a decrease in rRNA abundance (relative to total RNA) (*Figure 5C*) and the 28S/18S rRNA ratio (*Figure 5D*). This is consistent with a previous report that processing of the 28S rRNA is more sensitive to 5-FU than processing of 18S rRNA (*Kanamaru et al., 1986*). We also noticed the appearance of unprocessed rRNA precursors in animals treated with FUDR (*Figure 5—figure supplement 1*, arrow); however, the small magnitude of effect makes it difficult to evaluate significance of the phenomenon. These effects of FUDR on rRNA were also suppressed by addition of uracil.

In addition, we analyzed the effect of FUDR on RNA profile in adult control *glp-1(e2141)* animals that never experienced ARD (*Figure 5E*). We observed a trend toward reduction of total RNA and rRNA/total RNA ratio upon FUDR treatment, although these effects did not reach statistical significance (*Figure 5F,G*). These results are consistent with the much less pronounced effect of the drug on well-fed control animals (*Figure 1*). We did however observe a modest but statistically significant effect of FUDR on 28S/18S ratio in control animals (*Figure 5H*), suggesting that rRNA metabolism may still be affected by the drug in control animals.

## Discussion

The phenomenon of *C. elegans* ARD provides an attractive opportunity to investigate how functionality of chronologically old tissues can be maintained or improved. The work described here indicates that restoration of somatic tissues and normal lifespan upon diapause exit occurs even in the absence of a functional germline or GSCs, indicating that signals arising from GSCs are unlikely to be causative for these effects. Moreover, restoration of somatic tissues does not require somatic cell divisions, mitochondrial replication, or cell cycle activation. Instead, our data support a model that exit from ARD involves early nucleolar reactivation, followed by a burst of rRNA production and subsequent amplification of total cellular RNA. It seems likely that this is necessary for the synthesis of new cellular material which is important for restoration of youthful structure and function. The drug FUDR interferes with nucleolar function, rRNA processing, and total RNA production and prevents morphological and functional somatic recovery upon exit from ARD.

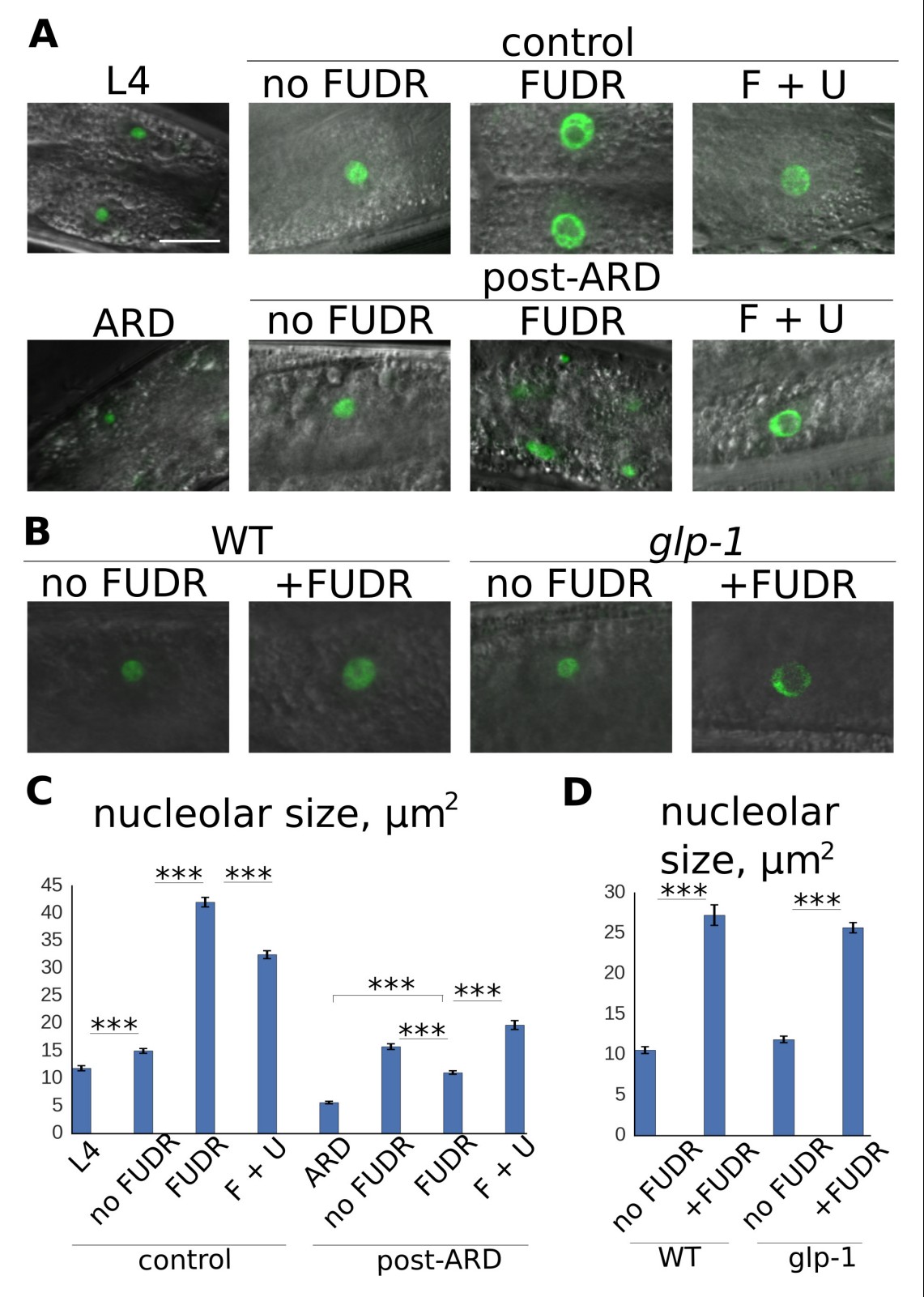

**Figure 4.** Post-ARD recovery is associated with nucleolar expansion. (**A**) Representative images of nucleolar morphology. Shown are nucleoli of intestinal cells visualized with fib-1::GFP. Control L4 and ARD animals were imaged before transfer to fresh OP50 plates containing no FUDR, 50 µM FUDR and 50 µM FUDR +1 mM uracil (F + U). Animals were then analyzed 4 days after transfer. Top panel: control animals at L4 stage and at 4 days of adulthood. Bottom panel: ARD animals before and after 4 days of recovery with or without FUDR. (**B**) FUDR induces nucleolar stress in post-mitotic

*Figure 4 continued on next page*

*Figure 4 continued*

tissues independently of the germline. Wild type and sterile *glp-1* mutants were grown at restrictive temperature (25°C) until day 1 of adulthood and then transferred onto FUDR-containing plates and analyzed 3 days later. (**C**) Quantification of nucleolar size from intestinal cells (in $\mu m^2$) of animals treated as in 4A. Shown are mean values ± SEM. See *Figure 4—source data 1*. (**D**) Quantification of nucleolar size from intestinal nucleoli (in $\mu m^2$) of animals treated as in 4B. Shown are mean values ± SEM. See *Figure 4—source data 2*. Also see *Figure 4—source data* . *$p < 0.05$; **$p < 0.005$; ***$p < 0.0005$. Scale bar is 10 $\mu m$.

DOI: https://doi.org/10.7554/eLife.36194.024

The following source data and figure supplement are available for figure 4:

**Source data 1.** Post-ARD nucleolar size.

DOI: https://doi.org/10.7554/eLife.36194.026

**Source data 2.**

DOI: https://doi.org/10.7554/eLife.36194.027

**Source data 3.** Nucleolar size, AL condition.

DOI: https://doi.org/10.7554/eLife.36194.028

**Figure supplement 1.** Change of nuclear and nucleolar sizes upon exit from ARD.

DOI: https://doi.org/10.7554/eLife.36194.025

Our findings point to rRNA production in particular as a key intracellular mediator of ARD maintenance and recovery. The nucleolus (*Gotta et al., 1997*; *Kennedy et al., 1997*) and rDNA (*Sinclair and Guarente, 1997*; *Kaeberlein et al., 1999*) were identified in early studies as key subcellular sites for longevity control in yeast, and more recent reports have indicated the importance of nucleolar plasticity in longevity of *C. elegans*, with smaller nucleoli being associated with longer lifespan (*Buchwalter and Hetzer, 2017*; *Tiku et al., 2017*). Consistent with this, we find that nucleoli are markedly smaller during ARD, while exit from ARD is associated with nucleolar expansion and amplification of the rRNA pool. In this regard, it is interesting that FUDR blocked the increase in rRNA upon exit from ARD but did not fully prevent expansion of the nucleolus, demonstrating that nucleolar size alone does not always reflect rRNA abundance and ribosome biogenesis. The nucleolus is a dynamic organelle that houses various enzymes for RNA processing and its composition is responsive to cellular stresses (*Boulon et al., 2010*). Therefore, one potential explanation for our results is that nucleolar size may be driven by early recruitment of rRNA-processing enzymes such as FIB-1 along with other machinery needed for ribosome biogenesis, and this step is not blocked by the drug. Alternatively, the increase of nucleolar size in post-diapause animals may reflect a compensatory response to impaired rRNA production caused by the drug, as we have observed that *ad libitum* fed control animals have larger nucleoli following treatment with FUDR.

A particularly striking feature of this study is the dramatic expansion of both rRNA and total RNA in post-ARD animals which is nearly completely prevented by treatment with FUDR. While this makes it difficult to pinpoint whether specific RNA species are most important for morphological and functional restoration post-ARD, we speculate that the decreased rRNA content and abnormal ratio of 28S/18S rRNA species are important, as these defects are likely to lead to impaired ribosome biogenesis and global mRNA translation. In particular, defects in ribosome biogenesis could drive the failure to expand the total mRNA due to reduced translation of transcriptional machinery. We also note that FUDR had a much smaller impact on the RNA content of somatic tissues in fully developed *ad libitum* fed adult animals, causing a trend toward reduced total RNA content and rRNA/total RNA ratio and a modest but significant reduction in 28S/18S rRNA ratio. These less pronounced effects of FUDR likely reflect that fact that somatic cells in adult animals are not undergoing the rapid growth and remodeling associated with exit from ARD, and therefore do not require as much new ribosome biogenesis and total mRNA production. Further studies will be needed to test these ideas.

Based on our observations, we speculate that normal longevity and post-diapause recovery may be regulated by distinct mechanisms. Metabolic attenuation, small nucleolar size, and stress resistance, perhaps similar to that associated with dietary restriction, are important both for survival during ARD as well as for long adult lifespan during normal aging. In contrast, somatic recovery and growth upon exit from ARD are associated with a rapid metabolic burst and associated amplification of rRNA and total RNA to support new protein synthesis. In this regard, it is interesting to note that

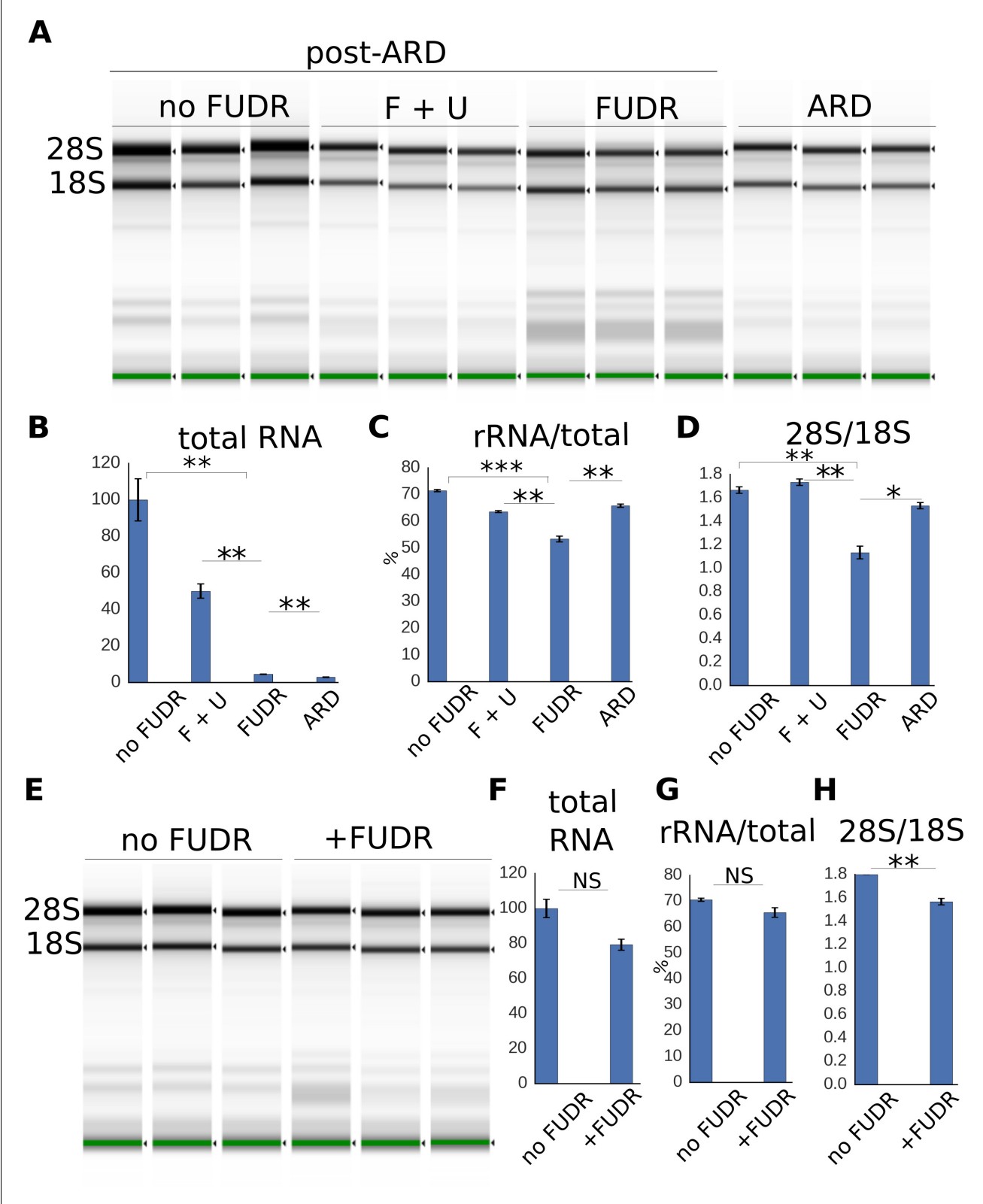

**Figure 5.** Expansion of the RNA pool during post-ARD recovery. (**A**) Total RNA extracted from post-ARD animals recovering in various conditions: no FUDR, 50 µM FUDR +1 mM uracil (F + U), 50 µM FUDR and ARD animals before recovery. Around 500 animals were used for no FUDR and FUDR +uracil sample preps, 3.5-fold more animals (around 1750) were used for FUDR and ARD preps. F + U and no FUDR samples were diluted 5-fold more than FUDR and ARD samples to fit into the quantitative range of TapeStation System. (**B,C,D**) Quantification of conditions shown in 5A.

*Figure 5 continued on next page*

Figure 5 continued

Normalized total RNA content (**B**), fraction of rRNA species in the total analyzed RNA, % (**C**) and ratio of 28S to 18S rRNA abundances (**D**). Shown are mean values ± SEM. Total RNA content was calculated from the TapeStation and adjusted to reflect dilution factor and number of animals (total RNA per animal). Values are expressed as % of RNA content in no FUDR sample, which was set as 100%. See **Figure 5—source data 1**. (**E**) Total RNA extracted from control animals that were transferred on fresh OP50 plates with or without FUDR at L4 stage. Around 300 animals were used per prep. (**F, G, H**) Quantification of conditions shown in 5E. Total RNA content (**F**), fraction of rRNA species in the total analyzed RNA, % (**G**), ratio of 28S to 18S rRNA abundances (**H**). Total RNA content is presented similarly to **Figure 5B**. Values are expressed as % of RNA content in no FUDR sample, which is set as 100%. All values are means ± SEM. See **Figure 5—source data 1**. *p<0.05; **p<0.005; ***p<0.0005.

DOI: https://doi.org/10.7554/eLife.36194.029

The following source data and figure supplement are available for figure 5:

**Source data 1.** RNA analysis.

DOI: https://doi.org/10.7554/eLife.36194.031

**Figure supplement 1.** Representative TapeStation electropherograms of total RNA extracted from ARD and post-ARD animals.

DOI: https://doi.org/10.7554/eLife.36194.030

intermittent fasting regimens where adult animals are subjected to periods of fasting intermixed with periods of *ad libitum* feeding have been found to increase lifespan in *C. elegans* (**Uno et al., 2013**; **Honjoh et al., 2009**) and mice (**Mattson et al., 2017**; **Xie et al., 2017**). It will be of interest to determine whether these types of regimens induce repeated cycles of nucleolar activation and contraction at the subcellular level and, if so, how this relates to metabolic activation and cellular restoration.

Although we are beginning to understand the cell-autonomous mechanisms of post-diapause recovery, the early signals that induce exit from ARD remain largely unknown. Likewise, the impact of specific tissues on whole-animal ARD recovery remains to be determined. For example, it may be that functional restoration of the pharynx and intestine are necessary for sufficient nutrient uptake to support restoration of the rest of the animal. The data presented here do demonstrate that successful transduction of this signal does not require a functional germline. Experimentally, exit from ARD is accomplished by providing the worms with bacterial food, so presumably food sensory pathways are transducing a pro-growth recovery signal from sensory neurons to somatic cells. Prior work has implicated both soluble and volatile food cues (**Smith et al., 2008**) along with neuronal serotonergic signaling in cell non-autonomous pro-longevity mechanisms (**Leiser et al., 2015**; **Petrascheck et al., 2009**; **Rangaraju et al., 2015**). It will be of interest to determine whether similar pathways become engaged during exit from ARD to promote growth and restore somatic tissues.

Finally, our work may shed light on the molecular mechanism of action of FUDR and, in particular, the inconsistent effects of FUDR on *C. elegans* lifespan and healthspan, as reported by a growing number of studies (**Van Raamsdonk and Hekimi, 2011**; **Aitlhadj and Stürzenbaum, 2010**; **García-González et al., 2017**; **Kato et al., 2017**; **Anderson et al., 2016**; **Angeli et al., 2013**). It has been assumed that these effects were the result of FUDR inhibiting DNA synthesis and reproduction, but our results point to a potentially important role for inhibition of RNA metabolism, including maturation of rRNA. Inhibition of RNA production by FUDR could exert broad effects on protein synthesis and cellular metabolism, particularly in cells or tissues undergoing growth or remodeling, or in genetic backgrounds that are particularly sensitive to perturbations in ribosome function or RNA metabolism. These effects should be considered, and when possible quantified, in future studies using FUDR.

It is of particular interest that FUDR prevented the functional and morphological restoration of post-ARD animals while having no detectable effect on restoration of full adult lifespan. This observation suggests that FUDR somehow uncouples healthspan from lifespan in post-ARD animals. The mechanism behind this uncoupling remains mysterious, but may provide a useful model for future studies aimed at understanding the mechanistic relationships between lifespan and healthspan. Such an understanding may be critical for development and implementation of interventions aimed at reducing morbidity and mortality in the aging populations.

# Materials and methods

**Key resources table**

| Reagent type (species) or resource | Designation | Source or reference | Identifiers | Additional information |
|---|---|---|---|---|
| Material and me   strain, strain background (*C. elegans*) | glp-1 (e2141); glp-1 | CGC | CGC: CB4037 | |
| Strain, strain background (*C. elegans*) | | CGC | CGC: DR2078 | |
| Strain, strain background (*C. elegans*) | | CGC | CGC: VC1224 | |
| Strain, strain background (*C. elegans*) | polg-1 (ok1548); polg-1 (ok1548/+) | This study | | Cross of DR2078 and VC1224 |
| Strain, strain background (*C. elegans*) | fib-1::GFP | CGC | CGC: COP262 | |
| S*train, strain background* (*C. elegans*) | tyms-1 | CGC | CGC: MJ65 | |
| Strain, strain background (*C. elegans*) | cye-1; cye-1 (eh10) | CGC | CGC: KM166 | |
| Strain, strain background (*E. coli*) | MG1693 (Δthy) | E. coli Genetic Resources at Yale | CGSC#: 6411 | |
| Strain, strain background (*E. coli*) | parent strain; control diet; BW25113 | E. coli Genetic Resources at Yale | CGSC#: 7636 | |
| Strain, strain background (*E. coli*) | Δtdk; JW1226-1 | E. coli Genetic Resources at Yale | CGSC#: 9112 | |
| Strain, strain background (*E. coli*) | Δupp; JW2483-1 | E. coli Genetic Resources at Yale | CGSC#: 9982 | |
| Chemical compound, drug | 5-Ethynyl-2'-deoxyuridine, EdU | Invitrogen | ThermoFisher: C10339 | |
| Chemical compound, drug | Uracil, U | Alfa Aesar | Fisher scientific: AAA1557018 | |
| Chemical compound, drug | Uridine, Urd | Alfa Aesar | Fisher scientific: A1522706 | |
| Chemical compound, drug | Thymine, T | Acros Organics | Fisher scientific: AC157850050 | |
| Chemical compound, drugMaterial and me | Thymidine, Thd | Alfa Aesar | Fisher scientific: A1149306 | |
| Chemical compound, drug | Deoxyuridine, dUrd | Alfa Aesar | Fisher scientific: A1602603 | |
| Commercial assay or kit | High Sensitivity RNA ScreenTape Ladder | Agilent | Agilent: 5067–5581 | |
| Commercial assay or kit | High Sensitivity RNA ScreenTape Sample Buffer | Agilent | Agilent: 5067–5580 | |
| Commercial assay or kit | High Sensitive RNA ScreenTape | Agilent | Agilent: 5067–5579 | |
| Sequence-based reagent | nd1 forward | IDT | | AGCGTCATTTATTG GGAAGAAGAC |
| Sequence-based reagent | nd1 reverse | IDT | | AAGCTTGTGCTAAT CCCATAAATGT |

## *C. elegans* strains and maintenance conditions.

The following strains were used in this study: N2, CB4037 (*glp-1 (e2141)*), DR2078 (*mIn1 [dpy-10 (e128) mIs14]/bli-2(e768) unc-4(e120) II*), VC1224 (*Y57A10A.15(ok1548)/mT1 II; +/mT1 [dpy-10 (e128)] II*), COP262 (*knuSi221 [fib-1p::fib-1(genomic)::eGFP::fib-1 3' UTR + unc-119(+)]*), MJ65 (*tyms-1(hc65) I*), KM166 (*cye-1(eh10)*). All strains were obtained from CGC. VC1224 and DR2078 were mated to introduce GFP-marked balancer chromosome into VC1224 background. COP262 was mated with CB4037 to analyze nucleolar dynamics in sterile animals. Animals were maintained at

20°C on NGM plates seeded with live *E. coli* unless otherwise indicated. Experiments with *glp-1 (e2141)* mutants were performed at 25°C to prevent germline development.

## Induction of and recovery from adult reproductive diapause

Adult reproductive diapause was initiated as described previously (*Angelo and Van Gilst, 2009*) . Briefly, hypochlorite-synchronized animals were allowed to grow until early mid-L4 stage on live OP50 *E. coli*. Developmental stage was assessed by the growth of germline and vulva development (*Frézal and Félix, 2015*; *Van Raamsdonk and Hekimi, 2011*) using dissecting and DIC microscopes. Sterile *glp-1* mutants were staged by vulva development. Animals were then gently washed off the plates into 15 ml Falcon tubes using M9 buffer with 0.1% Tween 20 (M9-Tw). After short centrifugation at 400 g, supernatant was removed, and fresh M9-Tw was added. Washing was repeated at least five times after which animals were transferred onto non-seeded NGM plates (BD plates) for starvation. BD plates were prepared as described previously (*Angelo and Van Gilst, 2009*) but supplemented with 50 µg/ml of carbenicillin. Animals were maintained on BD plates at 20°C in most of the experiments. All strains were grown and maintained at 25°C in experiments with *glp-1* mutants. Plates were examined regularly to spot potential contamination in which case plates were discarded.

For recovery from ARD, animals were gently washed off the BD plates with M9-Tw and transferred onto fresh NGM plates seeded with live OP50 *E. coli* and containing appropriate additives (50 µM FUDR, 1 mM uracil, 1 mM thymine, 40 mM hydroxyurea, 1 mM uridine, 1 mM deoxyuridine, 1 mM thymidine). Non-adult animals (larvae born during starvation and L4 larvae that did not reach adulthood) were manually removed under a dissecting microscope. Remaining animals were used for subsequent experiments (morphological analysis, lifespans, microscopy, health measures). Animal was considered recovered from diapause if it displayed visual improvements compared to starved animals (body size, intestine coloration, germline growth (for non-sterile strains), motility). *Ad libitum* fed L4 controls were transferred onto identical plates at the same developmental point at which ARD was usually initiated.

We used strains from the Keio *E. coli* deletion collection to analyze effect of bacterial diet on FUDR toxicity during post-ARD recovery. BW25113 (parent strain of the collection), JW1226-1 (Δ*tdk*) and JW2483-1 (Δ*upp*) cells were grown in LB media overnight and seeded on NGM plates containing 25 µM FUDR. Plates were allowed to dry for 2 days before experiment. ARD animals were transferred on the plates and scored for recovery as in other conditions.

Three independent biological replicates were performed to assess post-ARD recovery in each examined condition. We typically analyzed few to several dozen animals per condition in one experiment. Also see supporting file with detailed numerical data. Bar graphs in the figures show mean ± SEM calculated from recovery rates in 3 biological replicates.

## Lifespan assay

For lifespan assays (*Sutphin and Kaeberlein, 2009*), animals were transferred onto fresh plates every 2–3 days until reproduction had ceased, and then upon necessity to prevent food depletion. When FUDR was used, animals were transferred as necessary to prevent food exhaustion. To score viability, immobile animals were gently probed with a platinum wire. Animals that crawled off the plate and did not return and animals that died from internal progeny hatching were excluded from analysis. We typically started with 3 plates (with 40–45 animals per plate) per condition. Three independent biological replicates were performed for each lifespan experiment. Figures show results of individual representative experiments.

## Pumping and thrashing rate measurements

To analyze pumping rate, animals were observed under a dissecting microscope for 15–30 s, and the number of terminal bulb movements was recorded. We typically scored 10 animals per condition and calculated mean values ± SEM. Values are expressed as pumping rate per minute. Bar graphs in the figures show mean ± SEM from individual representative experiments.

For thrashing assays, animals were carefully transferred into a drop of clean M9 and the number of body bends during a 15–30 s observation period was recorded. We typically scored 10 animals per condition and calculated mean values ± SEM. Values are expressed as rate per minute.

Three independent biological replicates were performed for each pumping and thrashing measures presented. In case of thrashing assay, figures show mean ± SEM calculated for combined pool of animals from all three experiments.

## Microscopy

DIC and fluorescence images were acquired using a Zeiss Axiovert Compound Microscope and a Zeiss LSM780 Confocal Microscope. Animals were placed on 1% agarose pads and anesthetized with 0.2% tricaine/0.02% tetramisole solution. To visualize the germline, animals were fixed for 20 min in ice cold EtOH and stained with 4',6'-diamidino-2-phenylindole hydrochloride (DAPI) in M9-Tw.

For nucleoli analysis, we specifically examined intestinal cells using a confocal microscope using 40 × 1.2 NA water immersion objective. We set up optical slice thickness to 1 micrometer. We scored between 30–50 nucleoli from 5 to 10 animals per group for average size determination (shown are means ± SEM). We measured nucleolar size using ImageJ ellipse and free hand selection tools. Three independent biological replicates were performed for analysis of nucleolar changes. Bar graphs in the figures show mean ± SEM from individual representative experiments.

For simultaneous analysis of nucleolar and nuclear sizes, ARD and post-ARD animals were fixed in EtOH and stained with DAPI in M9-Tw. Animals were then imaged with 2 μm optical slices. Two independent biological replicates were performed. Around 20 nuclei/nucleoli pairs were examined in each experiment. Bar graphs in the figures show mean ± SEM from a representative experiment.

For estimation of ploidy, DAPI-stained animals were imaged on Zeiss LSM780 Confocal Microscope with 2 μm optical slices and 2 μm steps between slices. For each intestinal nucleus its DAPI signals from several slices were summed up and normalized for DAPI signals of diploid somatic or mitotic germline cells lying at the approximately same depth. The resulting values are therefore expressed as diploid equivalents. Two independent biological replicates were performed. A total of around 10 intestinal nuclei from 5 animals were examined in each trial. Bar graphs in the figures show mean ± SEM from a representative experiment.

## EdU labeling

To visualize newly synthesized DNA, MG1693 (Δthy) cells were grown in LB media overnight. 4 ml of the overnight culture was inoculated into 100 ml of M9 media supplemented with 5 ml of 20% glucose, 400 μl of 1.25 mg/ml thiamine, 5 μl of 10 mM thymidine, 100 μl 1M MgSO4, 100 μl 20 mM 5-ethynyl-2'-deoxyuridine (EdU). Cells were grown for additional 24 hr, concentrated by centrifugation and seeded on M9 agar plates. Cells were allowed to dry for 2 days and then used for experiments or stored at 4°C. Control animals were grown on EdU-labeled cells from egg till day 2 of adulthood. ARD animals were transferred on EdU-labeled cells for recovery and analyzed 5 days later.

For imaging, animals were washed in M9-Tw and fixed in ice-cold methanol. Click-labeling with Alexa-594 was then performed following manufacturer protocol (Invitrogen). Animals were also stained with DAPI. DAPI/Alexa-labeled animals were then examined using Zeiss LSM780 Confocal Microscope. We have performed two independent biological replicates with EdU labeling.

## Determination of mitochondrial DNA copy number

To measure mtDNA content we followed previously described qPCR approach (*Bratic et al., 2009*; *Rooney et al., 2015*). Briefly, individual *glp-1 (e2141)* animals (L1 larvae, ARD and post-ARD) were lysed by proteinase K digestion in 10 μL lysis solution (1 mg/ml proteinase K, 30 mM Tris pH8.0) for 25 min at 60°C. After inactivation of the proteinase K at 95°C samples were diluted up to 20 μL and used for qPCR reaction for NADH dehydrogenase subunit 1 (nd1) as described before (*Bratic et al., 2009*; *Rooney et al., 2015*). Three independent biological replicates were performed for qPCR measurements. In each experiment we analyzed 8 individual animals per condition with 3 technical replicates for each sample. Samples that did not amplify were not included into the analysis. Bar graphs in the figures show mean ± SEM from a representative experiment.

## Total RNA analysis

Age-synchronized adult animals were washed at least six times in M9-Tw and then incubated in the same buffer for about 20 min to allow clearance of gut bacteria. Animals were washed one more

time after that and transferred to TRIzol solution (Invitrogen). Total RNA was extracted using a Qiagen RNeasy kit following manufacturer recommendations. RNA concentration was first measured on Thermo NanoDrop spectrophotometer, and sample concentrations were adjusted to fit into quantitative range of the Agilent TapeStation. Total RNA, rRNA content, as well as RNA profiles were then analyzed using the Agilent 4200 TapeStation System using High Sensitive RNA ScreenTape following manufacturer protocols. Approximately 500 post-ARD animals per group were used in no FUDR and FUDR +uracil conditions, and 3.5-fold more animals were used in post-ARD FUDR and ARD conditions. We used approximately 300 animals per prep when analyzing the effect of FUDR on control animals. Total RNA content is normalized for the number of animals used in each prep and expressed as % of RNA content in the control (FUDR-free) sample, which is set as 100%.

Three independent biological replicates were performed with ARD animals. Two independent biological replicates were performed with control animals. Bar graphs in the figures show mean ± SEM from individual representative experiments.

## Statistical analysis

All the quantitative data were presented as means ± SEM. We performed comparison between groups using student t-test. P values less than 0.05 were considered significant (marked by * on bar graphs). P values less than 0.005 and 0.0005 were marked by ** and *** respectively.

## Acknowledgements

This work was supported by NIH grant R01AG031108 to MK and R00AG045341 to AM. NB was supported by NIA training grant T32AG000057.

## Additional information

### Competing interests

Matt Kaeberlein: Reviewing editor, *eLife*. The other authors declare that no competing interests exist.

### Funding

| Funder | Grant reference number | Author |
| --- | --- | --- |
| National Institute on Aging | T32AG000057 | Nikolay Burnaevskiy |
| National Institute on Aging | R01AG031108 | Matt Kaeberlein |
| National Institute on Aging | R00AG045341 | Alexander Mendenhall |

The funders had no role in study design, data collection and interpretation, or the decision to submit the work for publication.

### Author contributions

Nikolay Burnaevskiy, Conceptualization, Formal analysis, Investigation, Visualization, Methodology, Writing—original draft; Shengying Chen, Miguel Mailig, Anthony Reynolds, Shruti Karanth, Formal analysis, Investigation, Writing—review and editing; Alexander Mendenhall, Supervision, Visualization, Writing—review and editing; Marc Van Gilst, Conceptualization, Supervision, Funding acquisition, Writing—review and editing; Matt Kaeberlein, Conceptualization, Formal analysis, Supervision, Funding acquisition, Writing—original draft

### Author ORCIDs

Nikolay Burnaevskiy (iD) http://orcid.org/0000-0003-1885-8999
Alexander Mendenhall (iD) http://orcid.org/0000-0002-4716-7671
Matt Kaeberlein (iD) http://orcid.org/0000-0002-1311-3421

Decision letter and Author response
Decision letter https://doi.org/10.7554/eLife.36194.034
Author response https://doi.org/10.7554/eLife.36194.035

## Additional files

### Supplementary files
• Transparent reporting form
DOI: https://doi.org/10.7554/eLife.36194.032

### Data availability
All data generated or analysed during this study are included in the manuscript and supporting files.

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
