## [Decision Letter]

Thank you for submitting your article "Reactivation of somatic RNA metabolism mediates morphological rejuvenation after *C. elegans* adult reproductive diapause" for consideration by *eLife*. Your article has been reviewed by three peer reviewers, and the evaluation has been overseen by a Reviewing Editor and Kevin Struhl as the Senior Editor. The following individual involved in review of your submission has agreed to reveal his identity: Scott Kennedy (Reviewer #1).

The reviewers have discussed the reviews with one another and the Reviewing Editor has drafted this decision to help you prepare a revised submission.

Below are the reviews presented first, followed by the reviewer discussion, where the reviewers had other suggestions. As you can see, there is ample support for publishing this paper but some simple suggested experiments. The reviewers were mindful to not request inordinately laborious experiments or experiments that would constitute the next paper in this analysis; their suggestions were reasonable.

Reviewer #1:

In this manuscript by Burnaevskiy et al., the authors explore how *C. elegans* recover from starvation-induced adult reproductive diapause (ARD). The story begins with the observation that the nucleotide analog FUDR prevents recovery from ARD. Genetic studies suggest that the role of FUDR in ARD recovery is independent of the germline. Additionally, the authors find that, during ARD recovery, RNA metabolism is activated (~20x). Surprisingly, FUDR blocks this activation. Finally, the authors show that the effects of FUDR on ARD recovery (as well as RNA activation) can be suppressed by uracil, suggesting that RNA activation is required for ARD recovery and that the effects of FUDR on ADR are caused by FUDR's ability to prevent RNA activation. The methodology used in the paper seems sound and, for the most part, conclusions were justified by the data (see below). While it is not altogether surprising to me that RNA metabolism would need to be activated for animals to recover from ARD, the authors do a good job of establishing the point. In this regard, the uracil rescue experiment described in Figure 3D was particularly elegant and convincing. Additionally, it was quite surprising to me that FUDR would play such a pronounced role in RNA activation post-ARD, a factoid I suspect will be of interest to other researchers that use, or have used, FUDR in their experiments. For these reasons, I recommend publishing the work in *eLife*. Below, I list some suggestions that may improve the manuscript.

1) Attempts to link the ideas of tissue rejuvenation after ARD and tissue rejuvenation after aging felt shoehorned to me. I would guess that recovery from acute starvation and recovery from the damage of long-term aging are going to end up being different processes. I recommend the authors take another approach while framing the story.

2) Data showing that FUDR prevents rRNA reactivation (Figure 5), but not nucleolar expansion (Figure 4), post ARD seem, at least superficially, to be contradictory. The authors should include a description in their discussion of this issue and why they feel the results are not problematic for their model.

3) I did not think the rRNA/total and 28S/18S data in Figure 5 added much to the story. Also, rRNA processing defects (5J, arrow) were quite subtle and not definitive. I suggest the data be removed.

4) After reading the paper, I wanted to know what effect FUDR might have on RNA populations in non-ARD worms. If RNA metabolism is similarly affected, how do adult worms survive on FUDR plates? If RNA metabolism is recalcitrant to FUDR in non-ARD animals, why and how? The subject might be worth mentioning in discussion.

5) Results paragraph two. The logic of this introduction paragraph was difficult to follow. A rewrite is needed.

6) If memory serves, *glp-1*(ts) animals do not actually lack all germ cells (they still have a small pool (~8) of mitotically dividing germ cells). If I'm right, authors should modify text accordingly.

7) Essential information is missing from figure legends. Genotypes need to be listed (i.e. Figure 2C) and the number of independent replicates (vs. technical replicates) of experiments should be listed. Additional details would be helpful- such as were the samples in 5J diluted similarly to samples in 5A?

Reviewer #2:

The Burnaevskiy et al. manuscript reports that worms exposed to FUDR, which is generally thought to inhibit DNA replication, is not able to rejuvenate after ARD (adult reproductive diapause). The mechanisms allowing worms to rejuvenate after a prolonged period in ARD are largely unknown, so the finding that FUDR blocks rejuvenation is new and interesting. The authors also demonstrated that germlineless mutants can rejuvenate from ARD (in somatic tissues), indicating rejuvenation does not depend on a germline signal, and is likely cell autonomous. The data and conclusions on the effect of FUDR and germline are supported by solid data. The authors then performed a series of experiments in an attempt to identify the mechanism whereby FUDR blocks rejuvenation. This later part is more problematic, which are detailed below.

1) The *plog-1* mutant data are intriguing, but the fact that the homozygous mutants develop into morphologically normal, but sterile adults indicates that the mutant worms must have inherited sufficient amount of PLOG-1 to sustain somatic development. This indicated that maternal PLOG-1 is sufficient to support somatic DNA replication. Because of this, the fact that the *plog-1* mutant can rejuvenate its somatic tissues after ARD cannot be interpreted to indicate rejuvenation can take place in the absence of mitochondrial DNA replication. It is possible that maternal PLOG-1 is sufficient to support mito DNA replication in somatic tissues during post-ARD rejuvenation.

The authors should recover dissected somatic tissues (after removal of dissected gonads) and quantify mitochondrial DNA (using qPCR) in ARD and post-ARD animals to assess whether mitochondrial DNA replication can take place.

2) The uracil and thymidine supplement experiments need additional controls.i) Uracil can in fact be converted into deoxyuracil, which can be incorporated into DNA. The authors should do a parallel supplement experiment using deoxyuracil. If they get no rescue with deoxyuracil, but only with uracil, then that would provide much stronger evidence of an effect on RNA metabolism, and not DNA.ii) The *tyms-1* mutant experiment appears to suggest thymidine is in excess, as the worms likely get them through their bacteria diet. In this is true, then the thymidine supplement experiment is not as meaningful, as thymidine is likely already in excess. The authors can use a bacterial strain lacking thymidine to further test this point.

3) Although the authors observed interesting changes in nucleolar size in worms treated with FUDR, ultimately the size of the nucleolar did not appear to correlate with rejuvenation / rRNA biogenesis.

First, FUDR did NOT block nucleolar expansion in post-ARD worms, even though these worms failed to rejuvenate. Also, FUDR induced greater nucleolar expansion in control wild-type worms, even though these worms did not need to rejuvenate.

Moreover, total RNA remained very low in post-ARD worms treated with FUDR. Total RNA levels were not significantly different in control worms treated with FUDR. Therefore, total RNA levels did NOT correlate with nucleolar expansion.

It is unclear why the authors spent three paragraphs in the Discussion stating the importance of nucleolar size / expansion, while there is little evidence that this change is important for the recovery of RNA levels / rejuvenation post-ARD. The Discussion needs to be rewritten to discuss the contradicting results on nucleolar size.

4) Figure 5 indicates that total RNA levels plummeted during ARD, and their levels recovered during rejuvenation. These results indicated that ALL or some RNAs could be important for rejuvenation. The authors hypothesized that rRNA biogenesis to be key, however the evidence to support this is weak, largely based on a skewed ratio of rRNAs recovery relative to total RNA (and an UNcorrelated nucleolar expansion). The authors need to use a genetic way to specifically interfere with rRNA processing / biogenesis, and test whether that is sufficient to block rejuvenation. This will be a key and necessary experiment to support the authors' conclusion that rRNA processing / biogenesis is important.

Reviewer #3:

*C. elegans* adult reproductive diapause (ARD) is a state of reproductive quiescence that forms in response to late larval food deprivation. Upon ARD entry, animals remodel much of their germline and other tissues, and live months without food. Upon ARD recovery, they rejuvenate germline and soma, and reproduce. This is therefore an important model for long-lived states of quiescence and rejuvenation. However, the genetic and physiologic requirements for entry, maintenance and exit are relatively unknown.

This study by Kaeberlein and colleagues explores some of the requirements for ARD recovery. They show that the nucleotide analog FUDR inhibits recovery, while having little effect on late larval development under ad libitum conditions. Similarly a temperature sensitive mutation in the thymidylate synthase, *tyms-1*, inhibits recovery. Interestingly, uracil supplementation restores recovery in the presence of FUDR, while thymine does not, implicating RNA rather than DNA metabolism.

Indeed they find that overall levels of RNA are down in ARD and stay low with FUDR treatment. The authors associate recovery to effects on rRNA production, since animals in ARD have small nucleoli and low levels of rRNA, while animals undergoing recovery enlarge their nucleoli and produce higher levels of rRNA. FUDR exposure in ARD inhibits rRNA production and may cause nucleolar stress.

This study addresses an important question on a poorly understood state of organismal quiescence and rejuvenation. The work implicates RNA rather than DNA metabolism as the major process important for recovery. The implication of RNA metabolism in the recovery process is intriguing, but perhaps not surprising, since global processes such as rRNA and protein synthesis would seem to be essential to initiate (post-mitotic) regrowth. One might find that cycloheximide treatment would similarly affect recovery. Yet the connection to rRNA production and rejuvenation is correlative and not proven, since FUDR could be arresting growth through other means, (e.g. nucleolar stress induced arrest through p53?) with secondary consequences on recovery. The authors also test the potential contribution of POLgamma, and germline proliferation to recovery but do not really explore other genetic requirements. While the direction of the authors work is intriguing, the study seems somewhat limited in scope and correlative.

Other points:

1) Can the authors rule out that ARD worms might be more sensitive to drugs or toxins in general?

2) Recovered ARD worms on FUDR pump little (similar to ARD worms, Figure 2). Perhaps they do not eat enough to start recovery. Can this be ruled out?

3) The authors should measure DNA amounts in the *glp-1* ARD worms and upon recovery to have a clear measure of DNA metabolism.

4) Quality of the pictures must be improved (e.g. orientation, spacing of the worms, illumination of the pictures, no air bubbles). Scale bars missing, graphs are of different sizes. Genes should be in italic.

5) Figures showing nucleoli should also show DAPI staining of the nucleus.

Nucleolar to nuclear ratios might be informative, since intestinal nuclei also undergo endoreplication.

6) It looks like FUDR treatment causes rRNA degradation. Might FUDR activate a degradative processes rather than inhibit synthesis?

Comments from the consultation:

Reviewer 1:

The deoxyuracil control experiment suggested by reviewer #2 seems like a reasonable request. I also agree with #2 that "rRNA explains ARD recovery" hypothesis is weak. Authors could do more experiments to test the idea as suggested, or just drop the claim. I am ok with either. Finally, discussing the disconnect between role of FUDR on nucleolar size and ARD recovery is definitely needed. I agree with #3 that the "RNA-metabolism required for ARD recovery" hypothesis is correlative and not actually all that surprising. I liked the paper, however, and with some editing think it will make a fairly interesting read.

Reviewer 2:

Out of the three lines of experiments I suggested, I also think the deoxyuracil is the most straight forward and can really help to strengthen their argument that RNA metabolism is key.

I'm not completely convinced by their argument that DNA replication is not involved – so I suggested a follow-up there. I think Reviewer 3 also suggested some follow up on testing DNA replication further?

I am OK with them not doing more experiments to better support the rRNA and nucleolar expansion model, if they would substantially dial down on their speculation in the Discussion.

Reviewer 3:

I would support this paper with revisions.

1) The disconnect between FUDr and nucleolar size could be handled in discussion.

2) The deoxyuracil feeding experiment would be good.

3) If they want to solidify the connection to rRNA they could knockdown the Pol1 subunit rrn-3, and see if this impacts recovery. Or they could deal with the issue as a point of discussion.

I would add that the figures need major improvement.

---

## [Author Response]

Below are the reviews presented first, followed by the reviewer discussion, where the reviewers had other suggestions. As you can see, there is ample support for publishing this paper but some simple suggested experiments. The reviewers were mindful to not request inordinately laborious experiments or experiments that would constitute the next paper in this analysis; their suggestions were reasonable.Reviewer #1:[…] 1) Attempts to link the ideas of tissue rejuvenation after ARD and tissue rejuvenation after aging felt shoehorned to me. I would guess that recovery from acute starvation and recovery from the damage of long-term aging are going to end up being different processes. I recommend the authors take another approach while framing the story.

We appreciate the reviewer’s perspective on this and we have toned down discussion of potential similarities between normal and “diapaused” aging and we made textual changes to highlight that we are interested in both preservation of tissue functionality by diapause and its restoration after diapause exit. We have also largely replaced the word “rejuvenation” with words like “restoration of function”, “improvement”, and “recovery” to describe diapause exit. We agree that the mechanisms of aging/damage that occur during normative aging could very well be different than those that occur during ARD. Having said that, we and others have seen evidence for at least superficially similar types of damage during ARD and other diapause states in worms, including structural degeneration in muscle, loss of proteostasis, and mitochondrial dysfunction. Thus, we believe it is an open question.

2) Data showing that FUDR prevents rRNA reactivation (Figure 5), but not nucleolar expansion (Figure 4), post ARD seem, at least superficially, to be contradictory. The authors should include a description in their discussion of this issue and why they feel the results are not problematic for their model.

Thank you for noting this potentially confusing aspect of the data. We believe that our observations indicate that nucleolar expansion occurs when the cell attempts to induce ribosome biogenesis, which is blocked by FUDR. We have attempted to clarify our interpretation in the Discussion with the following text:

“In this regard, it is interesting that FUDR blocked the increase in rRNA upon exit from ARD but did not fully prevent expansion of the nucleolus, indicating that nucleolar size alone does not always reflect rRNA and ribosome biogenesis. […] Alternatively, the increase of nucleolar size in post-diapause animals may reflect a compensatory response to impaired rRNA production caused by the drug, as we have observed that *ad libitum* fed control animals have larger nucleoli following treatment with FUDR.”

3) I did not think the rRNA/total and 28S/18S data in Figure 5 added much to the story. Also, rRNA processing defects (5J, arrow) were quite subtle and not definitive. I suggest the data be removed.

We can appreciate this viewpoint and we have attempted to tone down the emphasis on rRNA, although we feel this is the most likely explanation for the effects of FUDR in the context of restoration of function following ARD as well as the various phenotypes associated with FUDR treatment in otherwise normally maintained animals. In response to these concerns, we have moved panel 5J into supplemental images (Figure 5—figure supplement 1) and restrained ourselves from drawing strong conclusions from this result. We feel it is important to make this piece of data available to the scientific community, since it may provide useful information for other labs, considering the widespread use of this drug among *C. elegans* researchers.

We have additionally attempted to reshape the RNA part of the story according to common suggestions of the reviewers to tone down rRNA hypothesis. We present analysis of changes of RNA profile in post-ARD and control conditions and highlight the changes of rRNA content, but we do not insist on rRNA changes as the key mechanism.

4) After reading the paper, I wanted to know what effect FUDR might have on RNA populations in non-ARD worms. If RNA metabolism is similarly affected, how do adult worms survive on FUDR plates? If RNA metabolism is recalcitrant to FUDR in non-ARD animals, why and how? The subject might be worth mentioning in discussion.

We have added a brief discussion of this. The data in Figure 5 demonstrate that RNA metabolism is not affected as strongly in somatic tissues of non-ARD adult animals (these experiments were done in *glp-1* animals to avoid effects on the germline), which likely reflects the fact that adult *C. elegans* are not undergoing rapid growth and therefore likely do not need to produce as many new ribosomes or RNA. We do detect a trend toward decreasing total RNA content and rRNA/total RNA ratio in adult animals and 28S/18S was modestly but significantly affected in control (non-ARD) animals.

5) Results paragraph two. The logic of this introduction paragraph was difficult to follow. A rewrite is needed.

This paragraph has been revised for clarity.

6) If memory serves, glp-1(ts) animals do not actually lack all germ cells (they still have a small pool (~8) of mitotically dividing germ cells). If I'm right, authors should modify text accordingly.

Thank you for correcting this. We have revised the text to indicate that a small number of GSCs may still be present in the *glp-1(e2141)* mutants. Literature suggests between 0-12 GSCs persist.

7) Essential information is missing from figure legends. Genotypes need to be listed (i.e. Figure 2C) and the number of independent replicates (vs. technical replicates) of experiments should be listed. Additional details would be helpful- such as were the samples in 5J diluted similarly to samples in 5A?

We have added the missing information. We indicate the number of independent replicates in the Materials and methods and in the numerical data section.

Reviewer #2:[…] 1) The plog-1 mutant data are intriguing, but the fact that the homozygous mutants develop into morphologically normal, but sterile adults indicates that the mutant worms must have inherited sufficient amount of PLOG-1 to sustain somatic development. This indicated that maternal PLOG-1 is sufficient to support somatic DNA replication. Because of this, the fact that the plog-1 mutant can rejuvenate its somatic tissues after ARD cannot be interpreted to indicate rejuvenation can take place in the absence of mitochondrial DNA replication. It is possible that maternal PLOG-1 is sufficient to support mito DNA replication in somatic tissues during post-ARD rejuvenation.The authors should recover dissected somatic tissues (after removal of dissected gonads) and quantify mitochondrial DNA (using qPCR) in ARD and post-ARD animals to assess whether mitochondrial DNA replication can take place.

We thank the reviewer for this comment and apologize for not more clearly stating that prior work with this mutant indicated no active replication of mtDNA during development or adulthood suggesting that there already exist sufficient copies of the mtDNA following embryogenesis to support development (Bratic *et al.*). To further confirm this is the case, we performed single worm qPCR measurement of mitochondrial DNA content using the *glp-1* mutant strain. We did not detect mtDNA increase in post-diapause animals. The data are now available in the Figure 3—figure supplement 1B.

2) The uracil and thymidine supplement experiments need additional controls.i) Uracil can in fact be converted into deoxyuracil, which can be incorporated into DNA. The authors should do a parallel supplement experiment using deoxyuracil. If they get no rescue with deoxyuracil, but only with uracil, then that would provide much stronger evidence of an effect on RNA metabolism, and not DNA.

We believe that the reviewer meant deoxyuridine. We considered this experiment, but felt that it was not informative because deoxyuridine would likely compete with FUDR (which is 5-fluoro-deoxyuridine) for the same enzymatic target(s). Further deoxyuridine can be converted to uridine through the same mechanism as FUDR is converted to 5-fluorouridine, potentially diluting out any toxicity from 5-fluorouridine. Thus, we reasoned that deoxyuridine will rescue the effects of FUDR regardless of whether FUDR is impairing DNA synthesis, RNA synthesis, or both. Nonetheless, we have added this experiment in the revision as requested.

We have also added what we hope will be more convincing additional studies using uridine and thymidine. While the deoxyuridine results are difficult to interpret as described above, we observed that uridine rescued phenotypes associated with FUDR while thymidine did not, arguing that RNA and not DNA metabolism are essential during post-ARD recovery. The data are now available in the Figure 3—figure supplement 1C,D.

To further test this model, we also performed additional experiments using bacterial strains deficient in nucleotide metabolism as described in the next response.

ii) The tyms-1 mutant experiment appears to suggest thymidine is in excess, as the worms likely get them through their bacteria diet. In this is true, then the thymidine supplement experiment is not as meaningful, as thymidine is likely already in excess. The authors can use a bacterial strain lacking thymidine to further test this point.

Thank you for suggesting the use of bacterial strains deficient in nucleotide metabolism. We chose not to utilize thymidine-deficient bacterial cells because these cells do not grow without thymine and succumb to “thymidineless death”. We reasoned that reduced growth and viability of the bacterial food source could prevent interpretation of this experiment.

We have therefore performed an alternative experiment that would be easier to interpret. It has been shown recently that bacterial metabolism affects response of *C. elegans* to drugs, including FUDR. We therefore used bacterial strains deficient in nucleotide-processing enzymes, namely upp (uracil phosphoribosyltransferase) (strain JW2483-1), tdk (thymidine kinase) (strain JW1226-1), and parent strain of the Keio collection BW25113. Cells were grown in LB and seeded on NGM plates containing FUDR. We predicted that upp-deficient strain should alleviate toxicity of FUDR, due to inability to metabolize the drug into RNA precursor. In contrast, we expected that tkd-deficient cells, which are unable to process FUDR into thymidine analog, should not reduce toxicity of FUDR compared to the parent strain. As shown in the new Figure 3C, this is exactly what we have found. These results further argue that RNA, and not DNA metabolism is essential for post-ARD recovery. The data are now available in the Figures 3C,F.

3) Although the authors observed interesting changes in nucleolar size in worms treated with FUDR, ultimately the size of the nucleolar did not appear to correlate with rejuvenation / rRNA biogenesis.First, FUDR did NOT block nucleolar expansion in post-ARD worms, even though these worms failed to rejuvenate. Also, FUDR induced greater nucleolar expansion in control wild-type worms, even though these worms did not need to rejuvenate.Moreover, total RNA remained very low in post-ARD worms treated with FUDR. Total RNA levels were not significantly different in control worms treated with FUDR. Therefore, total RNA levels did NOT correlate with nucleolar expansion.It is unclear why the authors spent three paragraphs in the Discussion stating the importance of nucleolar size / expansion, while there is little evidence that this change is important for the recovery of RNA levels / rejuvenation post-ARD. The Discussion needs to be rewritten to discuss the contradicting results on nucleolar size.

We apologize for not more clearly describing these results and why we think they are important. We have modified the Discussion to clarify our interpretation. We do feel that the dramatic effect of ARD on nucleolar size is relevant and informative, especially in the context of other literature linking nucleolar size to longevity, but size alone is not a clear indicator of rRNA and ribosome biogenesis, as our results with FUDR clearly indicate. The Discussion of this has been modified as follows:

“In this regard, it is interesting that FUDR blocked the increase in rRNA upon exit from ARD but did not fully prevent expansion of the nucleolus, indicating that nucleolar size alone does not always reflect rRNA and ribosome biogenesis. Nucleolus is a dynamic organelle that houses various enzymes for RNA processing and its composition is responsive to cellular stresses. Therefore, one potential explanation for these results is that nucleolar size may be driven by early recruitment of rRNA-processing enzymes such as FIB-1 along with other machinery needed for ribosome biogenesis, and this step is not blocked by the drug. Alternatively, the increase of nucleolar size in post-diapause animals may reflect a compensatory response to impaired rRNA production caused by the drug, as we have observed that *ad libitum* fed control animals have larger nucleoli following treatment with FUDR.”

4) Figure 5 indicates that total RNA levels plummeted during ARD, and their levels recovered during rejuvenation. These results indicated that ALL or some RNAs could be important for rejuvenation. The authors hypothesized that rRNA biogenesis to be key, however the evidence to support this is weak, largely based on a skewed ratio of rRNAs recovery relative to total RNA (and an UNcorrelated nucleolar expansion). The authors need to use a genetic way to specifically interfere with rRNA processing / biogenesis, and test whether that is sufficient to block rejuvenation. This will be a key and necessary experiment to support the authors' conclusion that rRNA processing / biogenesis is important.

We thank the reviewer for this comment, and we agree that we cannot definitively differentiated between the importance of total RNA, specific mRNAs, and ribosomal rRNA in this context. We respectfully disagree, however, that knocking down rRNA processing or biogenesis would be informative in this context. Genetically blocking rRNA processing and biogenesis would have a similar effect on total RNA to what we observed, due to the fact that functional ribosomes are required for production of transcriptional machinery during periods of robust growth. In addition, there is no experimentally clean way to perform such a study, due to the fact that RNAi feeding (to knock down rRNA processing in this case) upon release from ARD does not work efficiently (our unpublished data). We have now explicitly stated that we cannot rule out alternative models in the discussion and we hope this will satisfy this concern.

Reviewer #3:[…] Other points:1) Can the authors rule out that ARD worms might be more sensitive to drugs or toxins in general?

We have attempted to address this concern by treating ARD animals with hydroxyurea, which has several effects including inhibition of ribonucleotide reductase (DNA synthesis), catalase, and carbonic anhydrase. When ad libitum fed early-mid L4 animals are transferred on NGM media containing hydroxyurea, they experience rapid demise and collapse of body morphology. In contrast post-ARD animals manage to show some growth and visual improvements. Thus, for HU, post-ARD animals are more resistant to the drug. Therefore, it is unlikely that ARD animals are more sensitive to toxins in general. We have now added this result as Figure 1—figure supplement 1.

2) Recovered ARD worms on FUDR pump little (similar to ARD worms, Figure 2). Perhaps they do not eat enough to start recovery. Can this be ruled out?

This is a reasonable suggestion that we have now added to the discussion. Indeed, inability of the post-ARD animals to increase their pumping rate in the presence of FUDR may stall the rest of the recovery. Alternatively, inability of these animals to reinflate their intestine in the presence of FUDR may be another roadblock for the recovery. We cannot yet dissect the mechanistic role of different somatic tissues during recovery. Regardless of which tissue(s) become limiting for recovery, this does not impact our model that cell intrinsic effects on RNA metabolism underlie the deficits in restoration of function.

3) The authors should measure DNA amounts in the glp-1 ARD worms and upon recovery to have a clear measure of DNA metabolism.

Following suggestion from another reviewer we have performed qPCR on *glp-1* animals to measure mitochondrial DNA content. We do not detect increase of mtDNA content in post-diapause animals. See Figure 3—figure supplement 1B.

We have also added additional analysis of the nuclear DNA. We decided not to use qPCR to measure nuclear DNA content, because this assay likely would not be sensitive enough: ARD animals are anatomically mature and qPCR most likely would not detect a small burst of nuclear DNA synthesis, if let’s say few cells undergo final round of division and endoreduplication. We therefore applied alternative approaches to analyze the role of nuclear DNA metabolism during recovery.

First, we imaged newly synthesized DNA in post-ARD and control animals using nucleotide analog EdU with subsequent click-chemistry detection of the probe. We have observed birth of germline and somatic nuclei when control animals are grown in the presence of EdU from egg. In contrast, we can only detect EdU incorporation in the germline of post-ARD animals. These results indicate that no detectable somatic cell division occurs in post-ARD animals. We present the data in Figure 2D.

We also attempted to address the potential for DNA synthesis in endoreduplicating intestinal cells. EdU did not label endureduplication in the intestine even in control animals grown on EdU-labeled cells from egg. We therefore decided to better analyze intestine, since this tissue undergoes dramatic morphological improvement during recovery and intestinal cells usually undergo a final round of endoreduplication during adult molt. We estimated ploidy of intestinal cells in diapaused animals using DAPI staining. We have found that intestinal nuclei contain approximately 16 equivalents of diploid genomes, indicating that these cells finished endoreduplications. We show the data in the Figure 2E.

Third, we analyzed recovery of *cye-1* (cyclin E) mutant animals. These animals are maintained as a heterozygous strain with a balancer chromosome. Homozygous mutants are viable, but do not complete endoreduplications, have smaller body size and have clear defects in the development of the reproductive system. We analyzed recovery of these animals from the diapause and their sensitivity to FUDR. We found that animals do recover from the diapause normally (judged by pumping rate and overall body morphology) and are sensitive to FUDR. Therefore, reactivation of cell cycle (or endoreduplication) is not necessary for post-diapause recovery. The data are presented in the Figures 2F,G, H.

4) Quality of the pictures must be improved (e.g. orientation, spacing of the worms, illumination of the pictures, no air bubbles). Scale bars missing, graphs are of different sizes. Genes should be in italic.

We replaced low contrast images with better ones and added scale bars. We also changed the size of the graphs and italicized the genes.

5) Figures showing nucleoli should also show DAPI staining of the nucleus.Nucleolar to nuclear ratios might be informative, since intestinal nuclei also undergo endoreplication.

We have now added images of nucleoli with DAPI in ARD and post-ARD animals. We have also provided quantification of the nucleoli/nucleus ratio in these condition. See Figure 4—figure supplement 1. Also see Figure 3—figure supplement 1 for estimation of ploidy in ARD animals. As described above, our estimation of intestinal ploidy in ARD animals indicate that they did complete all the rounds of endoreduplication.

6) It looks like FUDR treatment causes rRNA degradation. Might FUDR activate a degradative processes rather than inhibit synthesis?

This is difficult to answer definitively based on our data or other limited data in the literature. The one piece of data arguing against significant degradation is that adult ad libetumfed animals show relatively modest changes in total RNA or rRNA in response to FUDR. We believe this reflects the much lower demand for new RNA synthesis in fully developed somatic tissues as opposed to rapidly growing somatic tissues undergoing remodeling and repair, such as upon release from ARD. It is however not unreasonable to suggest that if FUDR causes severe RNA stress then it can cause some RNA degradation as well. We have added this notion into the Discussion section.

Comments from the consultation:Reviewer 1:The deoxyuracil control experiment suggested by reviewer #2 seems like a reasonable request. I also agree with #2 that "rRNA explains ARD recovery" hypothesis is weak. Authors could do more experiments to test the idea as suggested, or just drop the claim. I am ok with either. Finally, discussing the disconnect between role of FUDR on nucleolar size and ARD recovery is definitely needed. I agree with #3 that the "RNA-metabolism required for ARD recovery" hypothesis is correlative and not actually all that surprising. I liked the paper, however, and with some editing think it will make a fairly interesting read.

1) We performed the requested deoxyuridine experiment as well as several additional experiments to address this point. Please see description of this and some other experiments above.

2) We toned down rRNA hypothesis and reshaped RNA part of the story.

3) We discuss potential reasons for disconnect between nucleolar size and post-ARD recovery.

Reviewer 2:Out of the three lines of experiments I suggested (Rev 2), I also think the deoxyuracil is the most straight forward and can really help to strengthen their argument that RNA metabolism is key.I'm not completely convinced by their argument that DNA replication is not involved – so I suggested a follow-up there. I think Reviewer 3 also suggested some follow up on testing DNA replication further?I am OK with them not doing more experiments to better support the rRNA and nucleolar expansion model, if they would substantially dial down on their speculation in the Discussion.

1) We did perform deoxyuridine experiment. Please see description of this and some other experiments above.

2) We have performed additional experiments to analyze DNA vs RNA requirements during recovery. Please see above.

3) We toned down rRNA hypothesis.

Reviewer 3:I would support this paper with revisions.1) The disconnect between FUDr and nucleolar size could be handled in discussion.2) The deoxyuracil feeding experiment would be good.3) If they want to solidify the connection to rRNA they could knockdown the Pol1 subunit rrn-3, and see if this impacts recovery. Or they could deal with the issue as a point of discussion.I would add that the figures need major improvement.

1) We toned down rRNA hypothesis and directly address the disconnect between nucleolar size and effects of FUDR.

2) We did perform deoxyuridine experiment. Please see description of this and some other experiments above.

3) Genetic inactivation of specific enzymes such as rrn-3 during recovery would be a great way to study post-diapause recovery. This approach would require inactivation of the candidate genes specifically during recovery, which in theory can be achieved through RNAi feeding. Unfortunately, our experience is that RNAi does not work efficiently in post-ARD animals. This is most likely due to the timing of RNAi effect, which may not be induced rapidly enough to prevent substantial signs of improvement in post-ARD animals during the first 48 hours of recovery.

4) We updated some of the microscopy images and we ensured similar sizes of bar graphs in different figures for a better aesthetic look.